# Rethinking Reasoning Quality in Large Language Models through Enhanced Chain-of-Thought via RL

## Abstract

Reinforcement learning (RL) has recently become the dominant paradigm for strengthening the reasoning abilities of large language models (LLMs). Yet the rule-based reward functions commonly used on mathematical or programming benchmarks assess only answer format and correctness, providing no signal as to whether the induced Chain-of-Thought (CoT) actually improves the answer. Furthermore, such task-specific training offers limited control over logical depth and therefore may fail to reveal a model's genuine reasoning capacity. We propose **D**ynamic **R**easoning **E**fficiency **R**eward (**DRER**) — a plug-and-play RL reward framework that reshapes both reward and advantage signals. (i) A *Reasoning Quality Reward* assigns fine-grained credit to those reasoning chains that demonstrably raise the likelihood of the correct answer, directly incentivising the trajectories with beneficial CoT tokens. (ii) A *Dynamic Length Advantage* decays the advantage of responses whose length deviates from a validation-derived threshold, stabilising training. To facilitate rigorous assessment, we also release *LogicTree*, a dynamically constructed deductive reasoning dataset that functions both as RL training data and as a comprehensive benchmark. Experiments show that DRER achieves significant improvements in reasoning accuracy and CoT quality over baseline methods across diverse training settings, while also reducing token usage during inference. Moreover, it demonstrates strong generalization on both reasoning and mathematical benchmarks, such as GPQA and AIME24. These results indicate that DRER, as a plug-and-play fine-grained RL reward framework, reliably strengthens reasoning behavior and provides a practical pathway toward enhancing the reasoning capabilities of large language models. All code and data are available in our anonymous repository `https://anonymous.4open.science/r/DRER-D34E`.

## 1 Introduction

Recent reasoning models (DeepMind, 2024; Qwen, 2024; Team et al., 2025), including R1-like reproductions (Team et al., 2025; Mei et al., 2025; Yu et al., 2025; Shao et al., 2024; Hu, 2025; Kool et al., 2019; Ahmadian et al., 2024; Sutton et al., 1998), have adopted reinforcement learning (RL) to enhance chain-of-thought reasoning. By systematically exploring verifiable reasoning paths that lead to correct answers, these methods incrementally boost performance and deliver remarkable gains. Current RL-driven CoT approaches typically train on mathematics and programming benchmarks (OpenAI, 2024; Guo et al., 2025; Cobbe et al., 2021; Chen et al., 2021), whose inherently stepwise solution procedures serve as natural proxies for logical inference (Wang et al., 2024a; Li et al., 2024), and they rely on rule-based reward (OpenAI, 2024; Guo et al., 2025) functions that assess only final answer correctness and formatting. This reliance stems from the straightforward evaluability of math and code tasks, where simple answer extraction or format checks suffice to assign reward signals and compute policy advantages.

However, this approach still faces two critical challenges. First, by relying solely on final-answer correctness as the reward signal, the model cannot distinguish which reasoning steps statistically boost the likelihood of the correct answers (Paul et al., 2024), nor quantify each token's substantive contribution to the conclusion; instead, it may lean on "decorative" chains that diverge from genuine

deductive paths (Zhang et al., 2024), thereby undermining the accurate evaluation and effective training of its reasoning ability.

Second, the corpora used to reinforce "reasoning ability" are almost entirely drawn from execution-verifiable domains (Sprague et al., 2024b)—such as mathematical problem sets and code synthesis tasks—while unified training data targeting pure formal logical inference remains severely lacking (Morishita et al., 2024). Such constrained training regimens risk conceptual overextension (Paul et al., 2024), whereby success on specific tasks is misconstrued as evidence of broadly applicable logical reasoning skills, potentially leading to an overestimation of the model's true inferential competence.

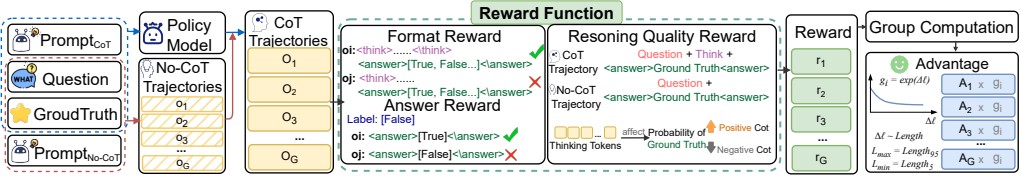

Figure 1: Overview of the Dynamic Reasoning Efficiency Reward (DRER) framework. $\text{Length}_{95}$ and $\text{Length}_5$ represent the 95th and 5th percentile lengths, respectively, computed from the validation set, and are used to normalize reasoning trajectory lengths according to task type or difficulty.

To address the limitations of outcome-only reward modeling in reasoning tasks, we propose *Dynamic Reasoning Efficiency Reward* (DRER), a plug-and-play reinforcement learning framework that reshapes both reward and advantage signals. DRER introduces two key mechanisms: (1) a *Reasoning-Quality Reward*, which assigns fine-grained credit to reasoning chains that statistically improve the likelihood of the correct answer, thereby reinforcing the utility of CoT tokens; and (2) a *Dynamic-Length Advantage*, which attenuates the policy advantage of responses whose lengths deviate from a validation-derived threshold, improving training stability. The overall framework is illustrated in Figure 1. In addition, we release *LogicTree*, a domain-agnostic deductive reasoning dataset carefully constructed to provide focused training supervision and to serve as a clean evaluation benchmark for identifying pathological reasoning behaviours.

Our experiments demonstrate that DRER significantly improves chain-of-thought (CoT) reasoning quality across different baseline algorithms and training corpora by providing fine-grained reward signals. When trained on the General Reasoning dataset, DRER consistently yields substantial improvements over both GRPO (Shao et al., 2024) and DAPO (Yu et al., 2025) base algorithms, achieving superior reasoning accuracy and CoT quality across a wide range of benchmarks. When trained on the LogicTree dataset, the combination of DRER and DAPO increases the accuracy of Qwen2.5-7B-Instruct-1M from $13.2\%$ to $60.0\%$, while reducing token consumption by approximately $75\%$ and achieving higher reasoning consistency. Taken together, these results show that DRER, as a plug-and-play fine-grained reward framework, reliably enhances the reasoning capabilities of LLM in diverse training settings and offers significant advantages over existing baseline methods.

The main contributions of this paper are summarized as follows:

- We propose **DRER** (**D**ynamic **R**easoning **E**fficiency **R**eward), a novel reinforcement learning rewawrd framework that adaptively reshapes both reward and advantage signals to improve CoT reasoning.
- We release *LogicTree*, a domain-agnostic benchmark for formal deductive reasoning that serves dual purposes: functioning as both a focused training set and a clean evaluation benchmark, while providing highlight insights into LLMs reasoning behaviours.
- We systematically validate our approach through extensive experiments, confirming the effectiveness of our methodology in improving both reasoning quality and efficiency.

## 2 PRELIMINARY

**Modeling Language Generation as a Token-Level MDP** Reinforcement learning aims to learn a policy that maximizes cumulative reward through interaction with an environment. We model language generation as a sequential decision process within a Markov Decision Process (MDP)

framework (Ouyang et al., 2022). Let $x = (x_0, \ldots, x_m)$ be the input prompt and $y = (y_0, \ldots, y_T)$ the generated response, with both drawn from a finite vocabulary $\mathcal{A}$. At step $t$, the state is $s_t = (x_0, \ldots, x_m, y_0, \ldots, y_t)$, and the action $a = y_{t+1} \in \mathcal{A}$ selects the next token. Transitions are deterministic: $\mathbb{P}(s_{t+1} \mid s_t, a) = 1$, where $s_{t+1} = (x_0, \ldots, x_m, y_0, \ldots, y_{t+1})$. Generation ends upon producing a terminal token $\omega$. The reward function $R(s, a)$ provides scalar feedback on output quality. The initial state $s_0$ is the tokenized prompt, sampled from a distribution $d_0$ over inputs. This MDP formulation allows reinforcement learning—both value-based and value-free—to align language model generation with desired objectives and human preferences.

**Group Relative Policy Optimization (GRPO)**  GRPO(Shao et al., 2024) removes the value function used in PPO(Schulman et al., 2017b) and estimates the advantages within a group of $G$ responses sampled by the behavior policy $\pi_{\theta_{old}}$ for each pair of questions-answers $(q, a)$. GRPO maximizes a PPO-style clipped objective with an explicit KL penalty:

$$\mathcal{J}_{\text{GRPO}}(\theta) = \mathbb{E}_{(q,a)\sim\mathcal{D},\,\{o_i\}\sim\pi_{\theta_{old}}}$$

$$\left[ \frac{1}{G} \sum_{i=1}^{G} \frac{1}{|o_i|} \sum_{t=1}^{|o_i|} \Big( \min\big(r_{i,t}\hat{A}_{i,t},\, \text{clip}(r_{i,t}, 1-\epsilon, 1+\epsilon)\hat{A}_{i,t}\big) - \beta\,\text{D}_{\text{KL}}(\pi_\theta \parallel \pi_{\text{ref}}) \Big) \right], \tag{1}$$

where

$$r_{i,t}(\theta) = \frac{\pi_\theta(o_{i,t} \mid q, o_{i,<t})}{\pi_{\theta_{old}}(o_{i,t} \mid q, o_{i,<t})}, \quad \hat{A}_{i,t} = \frac{R_i - \text{mean}(\{R_i\}_{i=1}^{G})}{\text{std}(\{R_i\}_{i=1}^{G})}. \tag{2}$$

GRPO first averages token-level losses within each response and then across the group, a sample-level aggregation that can implicitly favor longer responses and thus influence training dynamics (Liu et al., 2025).

**Decouple Clip and Dynamic Sampling Policy Optimization (DAPO)**  DAPO(Yu et al., 2025) shares GRPO's group-based sampling and advantage normalization, but differs in two key aspects. First, it replaces GRPO's symmetric clipping with asymmetric clipping bounds, allowing for unbalanced exploration and conservative updates. Second, it introduces a dynamic sampling constraint that requires both correct and incorrect responses in the sampled group to ensure meaningful advantage shaping. The resulting objective is:

$$\mathcal{J}_{\text{DAPO}}(\theta) = \mathbb{E}_{(q,a)\sim\mathcal{D},\,\{o_i\}\sim\pi_{\theta_{old}}}$$

$$\left[ \frac{1}{\sum_{i=1}^{G}|o_i|} \sum_{i=1}^{G} \sum_{t=1}^{|o_i|} \min\Big( r_{i,t}\hat{A}_{i,t},\, \text{clip}(r_{i,t}, 1-\varepsilon_{\text{low}}, 1+\varepsilon_{\text{high}})\hat{A}_{i,t} \Big) \right], \tag{3}$$

where optimization is applied only if the sampled responses are not all equivalent to the reference answer. $r_{i,t}$ and $\hat{A}_{i,t}$ are defined as in Equation 2.

**Reward Modeling**  Reward modeling in RL for LLMs is typically categorized into two approaches: rule-based rewards and learned reward models (RMs). Reward models, including outcome and process reward models (PRMs), learn a function through supervised learning, enabling finer-grained evaluation of intermediate reasoning steps. MATH-SHEPHERD (Wang et al., 2024b) and OmegaPRM (Luo et al., 2024a) show that PRMs improve reasoning consistency and generalization, but they also raise annotation costs, introduce potential data bias (e.g., MCTS-generated traces), and reduce reliability in early-step evaluation, which can destabilize training.

Rule-based rewards are more widely adopted, where simple criteria such as answer correctness and syntactic validity are used to evaluate model outputs. Representative works (Lyu et al., 2025; Xie et al., 2025; Li et al., 2025) like DeepSeek-R1 (Guo et al., 2025) utilize correctness-based signals to construct efficient and interpretable training pipelines. The primary advantages of rule-based rewards are twofold: firstly, they exhibit low implementation cost and, secondly, they are characterised by high transparency. These properties render them well-suited for large-scale RL training. However, their limitations are also evident: these methods only evaluate final outcomes, ignoring the quality of intermediate reasoning steps. As a result, models may learn to "shortcut" reasoning, producing correct answers without coherent or logically valid chains of thought—leading to misalignment between reasoning processes and outputs (Zhang et al., 2025).

# 3 METHOD

## 3.1 DRER

Rule-based rewards, such as answer correctness and format validity, minimal signals neglect to consider the reasoning trajectory that culminates in the ultimate response. Consequently, they may permit verbose, irrelevant chains of thought, which compromise reasoning transparency and reliability.

In order to address this limitation, a novel reward framework, Dynamic Reasoning Efficiency Reward (DRER), is introduced. This plug-and-play system has been designed to shape not only the correctness of final outputs, but also the efficiency and utility of intermediate reasoning steps.

Given an input question $x$, the large-language model (LLM) $\pi_\theta$ produces an output sequence $y$ autoregressively:

$$\pi_\theta(y \mid x) = \prod_{t=1}^{T} P_{\pi_\theta}(y_t \mid x, y_{<t}), \tag{4}$$

where the sequence $y = [c, a]$ denotes the model's output sequence, where the first contiguous segment $c = (c_1, \ldots, c_{T_c})$ comprises the CoT tokens and the second segment $a = (a_1, \ldots, a_{T_a})$ contains the answer tokens. The overall sequence length satisfies $T = T_c + T_a$.

We believe that if the generated CoT tokens $c$ are positive and coherent with the correct answer, it should *increase* the model's confidence in predicting ground-truth answer token:

$$\ell_{\text{CoT}} = \frac{1}{T_a} \sum_{t=1}^{T_a} \log \pi_\theta\big(a_t^\star \mid x_{CoT}, c, a_{<t}^\star\big), \quad \ell_{\text{NoCoT}} = \frac{1}{T_a} \sum_{t=1}^{T_a} \log \pi_\theta\big(a_t^\star \mid x_{NoCoT}, a_{<t}^\star\big), \tag{5}$$

CoT reasoning tokens that positively contribute to the model's ability to infer the correct answer should satisfy

$$\ell_{\text{CoT}} > \ell_{\text{NoCoT}}. \tag{6}$$

where $x_{CoT}$ and $x_{NoCoT}$ denote the CoT and no CoT input question respectively; $c = (c_1, \ldots, c_{T_c})$ is the generated CoT of length $T_c$; $a^\star = (a_1^\star, \ldots, a_{T_a}^\star)$ is the ground-truth answer consisting of $T_a$ tokens, and $a_{<t}^\star$ stands for its prefix up to position $t-1$; Finally, $\pi_\theta$ is the autoregressive language model policy parameterised by $\theta$.

To validate this hypothesis, we conduct experiments using Qwen2.5-7B-Instruct-1M on benchmarks. We first evaluate model-generated CoT trajectories using GPT-5.1 under a unified step-wise rubric, and examine how CoT quality scores correlate with the delta of log-probabilities. We then perform a CoT-disturbance test by comparing original CoT traces with shuffled and cross-question variants to assess delta of log-probabilities relevance to reasoning structure and semantic relevance. Finally, we analyze the impact of CoT on answer correctness by comparing likelihood shifts between CoT and no-CoT and examining fix and break rates across datasets. Full experimental details and results are provided in section 4.6.

**Reasoning Quality Reward**  To make the confidence-boosting property in equation 6 learnable, we define for each training instance $\mathbf{x}$ the log-likelihood margin

$$\Delta(\mathbf{x}) = \ell_{\text{CoT}} - \ell_{\text{NoCoT}}, \tag{7}$$

where $\ell_{\text{CoT}}$ and $\ell_{\text{NoCoT}}$ are given in equation 5. A positive $\Delta(\mathbf{x})$ indicates that the generated CoT reasoning tokens enhance the model's confidence in the correct answer, whereas a negative value reveals detrimental or spurious reasoning.

To obtain a numerically stable reward, we pass the margin through a smooth, bounded squashing function

$$R_q = \tanh\big(\Delta(\mathbf{x})\big), \tag{8}$$

yielding the *reasoning-quality reward*. The hyperbolic tangent preserves the sign of the margin, caps extreme values.

We incorporate $R_q$ into the overall reinforcement-learning objective by maximising the expected composite return

$$R = R_{\text{task}} + \lambda_q R_q, \tag{9}$$

where $R_{\text{task}}$ denotes the task-level reward (e.g., answer and format correctness) and $\lambda_q > 0$ is a weighting coefficient that balances task success and reasoning quality. This formulation directly rewards reasoning chains that demonstrably increase the likelihood of the correct answer while penalising uninformative or misleading chains, thereby systematically improving the model's logical reliability and interpretability.

**Dynamic Length Advantage** After every validation round we record the lengths $\{L_i\}$ of responses that are both correct and structurally valid within each difficulty bucket[1]. The empirical $5\%$ and $95\%$ quantiles define a dynamic lower and upper length bound, $L_{\min}^{(d)}$ and $L_{\max}^{(d)}$, respectively, for bucket $d$. For a training sample $i$ with effective response length $\ell_i$, we introduce a multiplicative attenuation coefficient

$$g_i = \exp\left(-\frac{\max\{0, L_{\min}^{(d)} - \ell_i, \ell_i - L_{\max}^{(d)}\}}{\tau}\right), \qquad \tau > 0, \tag{10}$$

where $L_{\min}^{(d)}$ denotes the 5th-percentile response length observed in the previous validation step for bucket $d$, while $L_{\max}^{(d)}$ corresponds to the 95th percentile in the same distribution. The variable $\ell_i$ represents the effective response length of the current sample $i$, and $\tau \in [5, 10]$ is a temperature hyperparameter that controls the decay rate of the attenuation function.

The attenuation is then applied to the advantage computed by GROUP COMPUTATION, $\hat{A}_i = g_i A_i$, so that responses that are excessively short ($\ell_i < L_{\min}^{(d)}$) or verbose ($\ell_i > L_{\max}^{(d)}$) are exponentially down-weighted. This mechanism penalises pathological length behaviours while preserving the signal of well-sized, high-quality chains of thought. The complete algorithm procedure of DRER is detailed in Appendix 1.

## 3.2 LOGICTREE

Most 'reasoning' benchmarks still fail to isolate formal deduction. Difficulty is inflated by injecting domain facts or arithmetic tricks, so logical skill is confounded with knowledge retrieval and calculation (Lin et al., 2025; Sprague et al., 2024b). Logical depth and structure remain almost uncontrollable: items rarely reveal how accuracy decays as inference chains lengthen, and no systematic consistency checks can be run across paraphrased versions of the same proof pattern (Saparov et al., 2023; Sprague et al., 2024a). Finally, intermediate steps are almost never evaluated; model capability is judged solely by the final answer (Paul et al., 2024).

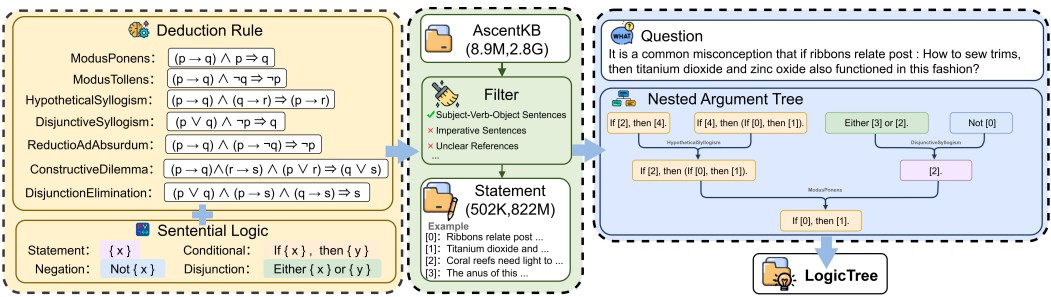

Figure 2: The framework of LOGICTREE automatic construction pipeline. We first sample atomic logic structures and sentences from seven deduction logic rules and four sentential logics, then fill it with natural statements in filtered AscentKB (Nguyen et al., 2021), and eventually construct the nested argument tree. Those intermediate will be hidden and transformed into questions.

Therefore, we present the LogicTree dataset, based on nested deductive reasoning rules that poses significant challenges to state-of-the-art LRMs. Solving these problems requires models to not only

---

[1] A bucket may correspond to a task type, question template, or any other granularity used in specific tasks.

recognize and correctly apply reasoning logic across diverse contexts but also to strategically plan hierarchical inference steps. Specifically, our dataset exhibits following key features:

**Programmatic Construction**, The reasoning depth, breadth, and number of sub-questions are fully controllable as shown in Figure 2 and Appendix A.2. Beyond evaluating models' judgment on root conclusions, intermediate reasoning steps are extracted and expanded into sub-questions. Compared to prior deductive reasoning benchmarks, this enables granular assessment of models' hierarchical reasoning accuracy.

**Diverse Logical Forms**, In contrast to grid puzzles or other logic games, LogicTree incorporates seven deductive reasoning rules and four sentential logic patterns, with each problem featuring distinct rule combinations. This significantly elevates the logical complexity.

**Probing LLMs' Foundational Reasoning**, We undertake multifaceted efforts to examine models' core logical capabilities. First, the dataset is decoupled from domain-specific knowledge to ensure models focus solely on pure logical reasoning. Second, we propose a logical consistency metric to evaluate models' ability to comprehend identical underlying logic across varying contextual representations.

Table 1: Deductive reasoning rules statistics on LogicTree 9.6k problems spanning depth 1-8.

| Deductive Rule | Logical Form | Amount |
|---|---|---|
| Modus Ponens | $(p \rightarrow q) \land p \implies q$ | 6 760 |
| Modus Tollens | $(p \rightarrow q) \land \neg q \implies \neg p$ | 6 750 |
| Hypothetical Syllogism | $(p \rightarrow q) \land (q \rightarrow r) \implies (p \rightarrow r)$ | 4 230 |
| Disjunctive Syllogism | $(p \lor q) \land \neg p \implies q$ | 6 865 |
| Reductio *ad Absurdum* | $(p \rightarrow q) \land (p \rightarrow \neg q) \implies \neg p$ | 6 780 |
| Constructive Dilemma | $(p \rightarrow q) \land (r \rightarrow s) \land (p \lor r) \implies (q \lor s)$ | 1 900 |
| Disjunction Elimination | $(p \lor q) \land (p \rightarrow s) \land (q \rightarrow s) \implies s$ | 6 625 |

**Evaluation** The LogicTree dataset is programmatically generated with full control over logical depth, sub-problem quantity, and reasoning variations, which enables multifaceted analysis of models' logic mechanism from novel perspectives.

We introduce three evaluation metrics: (1) Accuracy: Standard correctness rate, only credited when every sub-question is correctly answered; (2) Consistency Ratio: Reasoning stability across logically equivalent queries, measured as consistent correctness over several isomorphic questions; (3) $F\beta$-*Score*: Balances Answer Rate (proportion of valid *True/False* responses) and Precision (accuracy among valid responses) with parameter $\beta$.

Note that, unlike traditional NLI datasets with three-class classification (Cheng et al., 2025; Liu & Zhang, 2024) (*True*, *False*, or *Uncertain*), we restrict labels to *True/False* to mitigate semantic ambiguity that often artificially inflates accuracy by encouraging defaulting to *Uncertain*. LLMs may respond with *Unknown* during inference, reducing statistical noise from random guessing.

## 4 EXPERIMENT

### 4.1 EXPERIMENTAL SETTINGS

In the experiment section, we conduct 400-step post-training of the Qwen2.5-7B-Instruct-1M model using two baseline algorithms, DAPO and GRPO, with two distinct training datasets: LogicTree Data, focused on deductive logic, and General Reasoning Data, which blends mathematical and multi-domain reasoning data. This diversified training setup fully demonstrates the generality of the DRER framework. Specific experimental settings can be found in Appendix D.1. The main experiments evaluate the model on multiple public benchmarks and the LogicTree benchmark, confirming the enhancement of logical reasoning capability. Additionally, we perform detailed attribution and ablation studies to elucidate the mechanism and validate the effectiveness of each module within DRER.

## 4.2 MAIN RESULTS

**Training**  Throughout 400 training steps, we observe a monotonic rise in the model's accuracy on the LogicTree from 7% at the outset to nearly 60% in figure 5. Additionally, the reasoning steps are streamlined for greater conciseness and clarity. Detailed evaluation data are in Table 19. In both settings, DRER consistently improves final accuracy and accelerates convergence. Figure 3 and Figure 11 indicates the step at which the baseline (DAPO or GRPO) reaches its final precision, showing that DRER achieves a significantly higher or comparable performance earlier, highlighting its efficiency in guiding learning through structured reasoning signals.

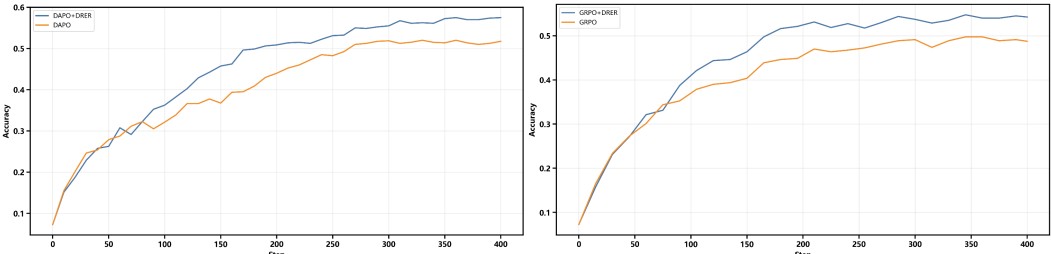

Figure 3: Accuracy on the LogicTree during post-training with DAPO (left) and GRPO (right), with and without DRER.

| Model | AIME 24 | MATH-500 | TheoremQA | MMLU-Pro | GPQA | LogiQA2.0 | ZebraLogic | LogicTree |
|---|---|---|---|---|---|---|---|---|
| Qwen2.5-7B | 12.8 | 55.8 | 21.1 | 38.8 | 27.9 | 45.7 | 30.9 | 13.2 |
| *Training on LogicTree Data* | | | | | | | | |
| GRPO | 13.1 | 54.7 | 18.2 | 38.4 | 27.1 | 47.1 | 33.5 | 45.1 |
| GRPO+DRER | 13.4 | **56.2** | **18.9** | 38.1 | 29.0 | **52.6** | **36.2** | 54.2 |
| DAPO | 13.9 | 55.9 | 17.6 | **40.1** | 33.5 | 46.5 | 32.3 | 52.4 |
| DAPO+DRER | **16.5** | 56.0 | 17.5 | 39.3 | **35.2** | 51.2 | 33.4 | **60.0** |
| *Traning on General Reasoning Data* | | | | | | | | |
| GRPO | 14.8 | 56.4 | 24.2 | 39.1 | 29.9 | 45.3 | 31.8 | 11.0 |
| GRPO+DRER | 17.2 | 59.2 | **25.1** | **39.7** | 35.4 | 46.7 | 31.6 | **14.1** |
| DAPO | 14.5 | 56.6 | 23.9 | 38.5 | 32.3 | 45.6 | 31.1 | 13.0 |
| DAPO+DRER | **18.3** | **61.8** | 22.8 | 39.0 | **38.6** | **47.5** | **32.4** | 12.1 |

Table 2: Performance on Mathematic and Reasoning benchmarks. Qwen2.5-7B model is referring to `Qwen/Qwen2.5-7B-Instruct-1M`. AIME24 and LogicTree results are reported as Avg@32 and Avg. score, respectively; all other datasets use standard accuracy.

Table 3: Comparison of Accuracy on LogicTree. For the complete results referring to Table 19.

| Model / Depth | 1 | 2 | 3 | 4 | 5 | 6 | 7 | 8 | Avg. |
|---|---|---|---|---|---|---|---|---|---|
| Qwen3-235B-A22B | **0.96** | **0.83** | 0.66 | **0.71** | 0.46 | 0.32 | 0.25 | 0.07 | 0.53 |
| Deepseek-R1 | 0.85 | 0.76 | 0.61 | 0.47 | 0.36 | 0.18 | 0.19 | 0.07 | 0.44 |
| Claude-3.7-Sonnet | 0.76 | 0.67 | 0.21 | 0.10 | 0.07 | 0.02 | 0.02 | 0.00 | 0.23 |
| GPT-o4-mini | 0.74 | 0.64 | 0.25 | 0.20 | 0.10 | 0.06 | 0.05 | 0.02 | 0.26 |
| GRPO | 0.81 | 0.71 | 0.58 | 0.42 | 0.45 | 0.20 | 0.20 | 0.11 | 0.45 |
| GRPO+DRER | 0.87 | 0.75 | 0.69 | 0.54 | 0.61 | 0.35 | 0.27 | 0.22 | 0.54 |
| DAPO | 0.88 | 0.73 | 0.66 | 0.47 | 0.60 | 0.36 | 0.23 | 0.20 | 0.52 |
| Ours (DAPO+DRER) | 0.90 | **0.83** | **0.76** | 0.59 | **0.67** | **0.45** | **0.31** | **0.31** | **0.60**$^{\uparrow 0.47}$ |

**Evaluation**  The main experimental findings are presented in Table 2, where we evaluated the trained 7B model under different training configurations across various benchmarks. Overall, our DRER framework, by performing fine-grained reward optimization on CoT tokens, consistently outperforms

baseline methods in eliciting the model's reasoning potential and enhancing its performance. Furthermore, it can be observed that when the model is trained exclusively on deductive reasoning data from LogicTree, it not only achieves notable improvements on LogiQA2.0 and ZebraLogic—both of which assess similar logical abilities—but also demonstrates generalization capability on mathematical benchmarks such as AIME24 and TheoremQA. When trained on General Reasoning, the model exhibits steady gains across mathematical benchmarks as well as multi-domain reasoning benchmarks including MMLU-Pro and GPQA, underscoring the broad applicability of the DRER framework.

For the detailed results on LogicTree, as demonstrated in Table 3, even advanced models such as GPT-o3-mini, DeepSeek-R1, and Claude3.7 achieve accuracy scores below $20\%$ across reasoning depths of 7-8 in LogicTree. The best performing model, Qwen3-235B, maintains the highest accuracy of $25\%$ on problems with reasoning depth of 7, with an average accuracy of $53\%$. This reveals significant deficiencies in the complex deductive reasoning capabilities of existing reasoning models. In contrast, our trained 7B model achieves state-of-the-art performance in terms of average accuracy, showing substantial improvement over the base model, and maintains a $31\%$ accuracy rate even at maximum reasoning depth.

Additionally, our experiments reveal distinct *Unknown* response tendencies across models. While GPT-o4-mini exhibits stronger reasoning capability than GPT-4o, their comparable accuracy stems from GPT-o4-mini's overcaution (excessive *Unknown* responses). However, GPT-o4-mini achieves significantly higher Precision and *Fβ-Score* scores in valid responses (details in Appendix 9).

### 4.3 ABLATION STUDY

To investigate the contribution of different design choices in DRER, we perform an ablation study.

**Effect of Reasoning Quality Reward.** We compare training runs with and without the Reasoning Quality Reward (RQR). As shown in Table 4, introducing RQR—which provides fine-grained credit assignment for CoT quality—leads to a substantial improvement in reasoning accuracy, whereas removing it results in a clear performance drop on both AIME24 and GPQA. Moreover, the training dynamics in Appendix Figure4 further corroborate this effect: the reasoning-quality reward steadily increases and eventually stabilizes at a high value during training, indicating that DRER consistently guides the policy toward CoT trajectories that enhance the model's confidence in the correct answer. Overall, RQR offers a precise and stable supervisory signal that enables the model to learn reasoning steps with genuine contribution, thereby improving both the quality of its reasoning process and the final prediction accuracy.

Table 4: Ablation experiment result on DRER. Compare the performance w/o Reasoning Quality Reward(RQR) or Dynamic Length Advantage(DLA). Avg@32 score is reported on AIME24.

| Method | AIME 24 | GPQA |
|---|---|---|
| DRER | **18.3** | **38.6** |
| w/o RQR | $14.7_{\downarrow 3.6}$ | $33.1_{\downarrow 5.5}$ |
| w/o DLA | $16.2_{\downarrow 2.1}$ | $35.3_{\downarrow 3.3}$ |

Table 5: Training dynamics of model response length w/o Dynamic Length Advantage(DLA).

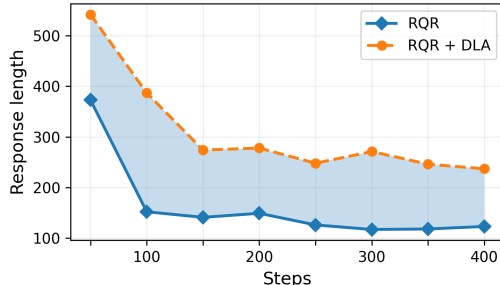

**Effect of Dynamic Length Advantage.** In the early stages of training, a small number of extreme-length responses can disproportionately influence the model's learned response-length distribution, leading to instability in optimization. Dynamic Length Advantage (DLA) mitigates this issue by applying advantage-level attenuation to such outlier trajectories, preventing them from dominating the learning dynamics. We compare training runs with and without DLA, and the results in Table4 and Table5 support its effectiveness. When DLA is removed, the model exhibits a slight drop in performance and shows substantially larger fluctuations in response length throughout training. These observations indicate that DLA effectively suppresses the destabilizing impact of extreme-length

Table 6: Comparison of Consistency Ratio on LogicTree. For the complete results referring to Table 20

| Model / Depth | 1 | 2 | 3 | 4 | 5 | 6 | 7 | 8 | Avg. |
|---|---|---|---|---|---|---|---|---|---|
| Qwen3-235B-A22B | **0.90** | 0.65 | 0.30 | **0.50** | 0.15 | 0.00 | 0.00 | 0.00 | 0.32 |
| Deepseek-R1 | 0.70 | 0.55 | 0.20 | 0.15 | 0.10 | 0.00 | 0.00 | 0.00 | 0.22 |
| Claude-3.7-Sonnet | 0.65 | 0.35 | 0.00 | 0.00 | 0.00 | 0.00 | 0.00 | 0.00 | 0.12 |
| GPT-o4-mini | 0.50 | 0.35 | 0.00 | 0.05 | 0.00 | 0.00 | 0.00 | 0.00 | 0.11 |
| GRPO | 0.55 | 0.50 | 0.40 | 0.25 | 0.45 | 0.20 | **0.15** | 0.00 | 0.29 |
| GRPO+DRER | 0.65 | 0.50 | 0.25 | 0.25 | 0.25 | 0.10 | 0.00 | 0.00 | 0.25 |
| DAPO | 0.65 | 0.45 | 0.45 | 0.20 | **0.50** | 0.10 | 0.05 | **0.10** | 0.31 |
| Ours (DAPO+DRER) | 0.70 | **0.70** | **0.60** | 0.35 | **0.50** | 0.35 | 0.05 | 0.10 | **0.41**$^{\uparrow 0.40}$ |

samples while allowing DRER to focus optimization on reasoning quality rather than being driven by pathological length patterns.

## 4.4 DOES MODEL REALLY LEARN THE LOGICAL PARADIGM?

A key question remains whether models truly understand logic or merely memorize puzzles. While prior work (Cheng et al., 2025) reveals models' tendency for self-contradiction on logically equivalent propositions, LogicTree naturally evaluates this through problems sharing identical logical structures but varying linguistic instantiations. Our Consistency Ratio metric quantifies this capability.

As shown in Table 6, most models can understand simple deductive reasoning logic, but at reasoning depths of 7-8, even state-of-the-art models such as GPT-o3-mini, Qwen3-235B, deepseek-r1, and Claude3.7 demonstrate consistency rates approaching zero, revealing current models' insufficient capability for consistent extended thinking and complex combinatorial logic.

Additionally, we analyzed whether models explicitly utilized certain deductive reasoning rules in their responses. Results in the Appendix provide word-frequency statistics and examples for GPT-o4-mini, DeepSeek-R1, Qwen3-235B, and our model, indicating a drop in explicit paradigm mentions with growing logical complexity and uneven competence across paradigms. Moreover, there exhibits varying capabilities across different logical paradigms. For example, DeepSeek-R1 responses most frequently reference "*Modus Tollens*", while "*Disjunction Elimination*" appears substantially less often. This disparity may stem from either the inherent complexity of the latter rule or inadequate exposure during pre-training. Our framework shows improved rule identification capacity with increasing response length and logical complexity.

## 4.5 WHY DOES RQR ACCURATELY MEASURE REASONING QUALITY?

**Information-theoretic interpretation.** In information theory, the mutual information between two random variables $Z$ and $Y$ is defined as

$$I(Z;Y) = \mathbb{E}_{z,y}\left[\log\frac{p(z,y)}{p(z)\,p(y)}\right] = \mathbb{E}_{z,y}\left[\log p(y \mid z) - \log p(y)\right]. \quad (11)$$

Then, the mutual information between the chain-of-thought $z$ and the correct answer $y^*$ conditioned on the input $x$ can be expressed as:

$$I(z;y^* \mid x) = \mathbb{E}_{z,y^*|x}\left[\log p(y^* \mid x, z) - \log p(y^* \mid x)\right]. \quad (12)$$

As shown in Eq.7 and Eq.8, the RQR can be viewed as a sample-based estimator of the conditional mutual information $I(z;y^* \mid x)$ between the CoT and the correct answer. This quantity measures the *information gain* contributed by CoT tokens toward predicting the correct answer.

To further validate whether RQR can faithfully measure the quality of chain-of-thought (CoT) reasoning, we design more complementary experiments in Appendix E.

**GPT-5.1–Based CoT Quality Scoring**   To assess whether RQR provides a meaningful estimate of reasoning quality, we conduct a series of controlled evaluations using GPT-5.1 as an external judge of Chain-of-thought quality. At a high level, we compare CoT trajectories generated by the base model, the DAPO-only model, and the DAPO+DRER model, and examine how GPT-5.1's CoT quality scores correlate with the learned RQR. The full evaluation protocol and scoring rubric are provided in the appendix E.1.

The results show a clear monotonic relationship: trajectories assigned higher quality scores by GPT-5.1 consistently obtain higher RQR values, confirming that RQR tracks genuine improvements in reasoning behavior. Moreover, DRER training produces a decisive shift toward higher-quality CoT, with substantially higher GPT-5.1 scores than those of both the base and DAPO-only models. These findings demonstrate that RQR not only reflects reasoning quality but also serves as an effective training signal that leads to stronger, more coherent CoT reasoning.

### 4.6 DOES MODEL'S REASONING BEHAVIOUR BECOME MORE EFFECTIVE?

To isolate the effect of explanatory CoT on answer confidence, We test Qwen2.5-7B-Instruct-1M on 500 GSM8K and 500 LogicTree problems, generating for each prompt a direct answer (No-CoT) and a step-by-step CoT. We mark a CoT as **effective** if the model is *incorrect* in the No-CoT setting but *correct* with CoT. We compute the log-probability gain of the ground-truth answer tokens $a_t^\star$ as $\ell_{\text{CoT}} - \ell_{\text{NoCoT}}$.

Samples are categorized into four groups based on answer correctness: **(WR)** wrong No-CoT / right CoT, **(RR)** right No-CoT / right CoT, **(WW)** wrong in both, and **(RW)** right No-CoT / wrong CoT. Statistics are reported in Tables 15 and 16.

We further split the data by the sign of $\Delta\ell$ (Tables 17 and 18). For positive $\Delta\ell$, the model shows a higher fix rate (proportion of WR is higher), with a significant increase in transitions from wrong to correct answer. For negative $\Delta\ell$, the break rate is higher and the fix rate lower, making transitions from correct to wrong more likely.

Figure 12 and Figure 13 show the prediction distribution for a difficulty-3 problem from 100 samples. Compared to the DAPO 400-step baseline, the DRER-trained policy produces a markedly sharper peak around the ground-truth answer, indicating that the learnt reasoning tokens help concentrate probability mass on the correct solution.

Finally, Figure 6 and Figure 7 show that DRER keeps the average response length stable at fewer tokens, saving tokens per problem relative to the baseline while achieving higher accuracy. This validates DRER's ability to simultaneously improve reasoning quality and reduce inference cost.

## 5   CONCLUSION AND FUTURE WORK

We propose DRER, a plug-and-play reinforcement learning framework that explicitly links the contribution of each reasoning step to the model's confidence in the final answer. By jointly optimizing the reasoning-quality reward and the dynamic-length advantage, DRER encourages the model to produce logically meaningful and length-efficient chains of thought. In addition, we introduce LogicTree, a programmatically constructed benchmark with controllable logical depth, designed for rigorous evaluation of deductive reasoning in LLMs.

Extensive experiments demonstrate that DRER significantly improves reasoning accuracy, reasoning quality, and training convergence over baseline methods, confirming that reinforcing high-quality reasoning signals enhances robustness and transferability of reasoning capabilities. These results validate the practical effectiveness of fine-grained CoT reward shaping and highlight LogicTree as a reliable diagnostic environment for analyzing reasoning mechanisms in LLMs.

We release all code and the complete LogicTree corpus to ensure transparency and reproducibility. Together, DRER and LogicTree provide a lightweight, theoretically grounded basis for reasoning-aligned RL, enabling safer and more interpretable LLMs in logic-critical domains. Future work should extend this framework to richer logics and multimodal data.

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

## A  TECHNICAL APPENDICES AND SUPPLEMENTARY MATERIAL

Table 7: An example of a logictree puzzle.

> **An example of a logictree puzzle**
>
> **Paragraph**:
> On the condition that coral reefs need light to grow so only occur in shallow waters, it is definitely the case that in addition to this, olive oil is also ideal for frying and is the most stable fat when heated.If in addition to this, olive oil is also ideal for frying and is the most stable fat when heated, then if ribbons relate post : How to sew trims, then titanium dioxide and zinc oxide also functioned in this fashion.It is a fact that either the anus of this invertebrate is located on top of its body or coral reefs need light to grow so only occur in shallow waters.The statement that 'the anus of this invertebrate is located on top of its body' is incorrect.
>
> **Question**:
> It is a common misconception that if ribbons relate post : How to sew trims, then titanium dioxide and zinc oxide also functioned in this fashion.
>
> **Solution**:
> False

### A.1  SEVEN DEDUCTIVE PARADIGMS IN LOGICTREE

LogicTree centres on seven classic deductive paradigms that constitute the atomic reasoning units of every sample. Each paradigm is implemented as a dedicated Python class (see `logic.py`) whose constructor generates the required premises and the logically entailed conclusion. The table below summarises their formal schemata together with bilingual surface examples.

### A.2  LOGICTREE: TEMPLATE AND CONSTRUCTION

We construct LogicTree through three automated steps:

1. **Logical Node Sampling.** Atomic premises and target conclusions are sampled from seven classical deductive rules (e.g., Modus Ponens, Modus Tollens) and four sentential logics, generating symbolic propositions.

2. **Natural-Language Instantiation.** Each symbolic proposition is mapped to natural declarative statements retrieved from the filtered AscentKB corpus Nguyen et al. (2021), excluding ambiguous expressions or compound sentences to enhance lexical diversity while maintaining clarity.

3. **Nested-Tree Assembly.** The instantiated nodes are recursively composed into reasoning trees with configurable depth and width. Intermediate conclusions are masked from given premises, then transformed into sub-questions to create multi-step problem instances. This design ensures the inference process depends solely on logical form rather than sentence semantics, effectively decoupling reasoning from world knowledge.

Table 8: Model Response of logictree.

---

**Model Response of logictree**

**Paragraph**:
When the notion that 'if the statement that states the worms also eat the food scraps and worm bin bedding is false, then the statement 'emergent wetland vegetation is rooted in soil that is under the water for most of the time' can be considered false' is untrue is true, it follows that hydrangeas need minimal care in well-drained, fertile soil, and are shade lovers.One may reasonably assume that if the notion that 'if the statement that states the worms also eat the food scraps and worm bin bedding is false, then the statement 'emergent wetland vegetation is rooted in soil that is under the water for most of the time' can be considered false' is untrue, then those that suggest hydrangeas need minimal care in well-drained, fertile soil, and are shade lovers are wrong.Assuming sometimes, cancer cells left behind after treatment eventually form new tumors, we know that if either eggs from common ducks require about 28 days to hatch or the island was originally inhabited by Ligures Ilvates, who gave it the ancient name Ilva, then if the statement that states the worms also eat the food scraps and worm bin bedding is false, then emergent wetland vegetation is rooted in soil that is under the water for most of the time.The statement that states sometimes, cancer cells left behind after treatment eventually form new tumors is right.Whenever the claim that either eggs from common ducks require about 28 days to hatch or the island was originally inhabited by Ligures Ilvates, who gave it the ancient name Ilva is false is true, the following statement is also true: the bowl acts as a carb hole.Whenever the claim that either eggs from common ducks require about 28 days to hatch or the island was originally inhabited by Ligures Ilvates, who gave it the ancient name Ilva is false is true, the following statement is also true: the statement 'the bowl acts as a carb hole' can be considered false.

**Question**:
The statement 'the worms also eat the food scraps and worm bin bedding' can be considered true.
It is a common misconception that if the statement that states the worms also eat the food scraps and worm bin bedding is false, then emergent wetland vegetation is rooted in soil that is under the water for most of the time.

**Model Response**:
1. Analyze the logical structure of the given statements.
2. Identify the conditions and implications in each statement.
3. Determine the truth value of the given statements and how they relate to each other.
4. Check the consistency of the statements with the provided premises.
5. Deduce the implications of the given statements on the conclusion.
6. Evaluate the final conclusion based on the logical consistency and the given premises.
```
</think>
<answer>[True, False]</answer>
```

**Solution**:
True
False

---

## A.3 PRIMITIVE AND COMPOUND PROPOSITIONS

LogicTree expresses every deductive instance in terms of one *primitive statement* and four *compound connectives*. The primitive `Statement` captures an atomic fact—e.g. "Alice studies."— while the four connectives build larger formulas: *negation*, *conjunction*, *implication*, and *inclusive disjunction*. Each connective is implemented as a dedicated class whose method `.nl()` randomly selects a surface template from `expressions.json`. Table 12 summarises the five constructs, their formal notation, and representative English renderings.

Table 9: Full Chain-of-Thought (CoT) Prompt Template Used for DRER Training and Evaluation.

---

**COT prompt**

**System Input**:
<|im_start|>system
You are a helpful assistant. The assistant first thinks step by step about the reasoning process in the mind and then provides the user with the answer.
The reasoning process and answer are enclosed within <think> . . . </think> and <answer> . . . </answer> tags, respectively, i.e.
<think> Write the reasoning process for the given paragraph here </think>
<answer> Fill in the final answer list for {num_q} question(s) here: True, False or Unknown. Like this: [True, False. . . ] </answer>
You must choose one of the following answers:
– TRUE: if the premises entail the statement
– FALSE: if the premises contradict the statement
– UNKNOWN: if you cannot determine the truth value of the statement from the premises
You will be given a paragraph of logical premises and a statement. Perform logical reasoning **strictly based on the premises** using propositional logic.
Assume all premises are true. Do not rely on prior world knowledge.
<|im_end|>

**User Input**:
<|im_start|>user
Paragraph: {paragraph}
{current_question}
<|im_end|> <|im_start|>assistant <think>

**Variable meanings**:
{num_q}: Number of questions in the current prompt.
{paragraph}: The paragraph containing the logical premises.
{current_question}: The specific statement whose truth value is to be evaluated.

---

**Lexicalization.** When generating a sample, the pipeline first creates `Statement` objects for the chosen entities, then composes them with the connectives above. For example, calling `Negation(S).nl()` yields a randomly chosen negated template such as *"The claim that S is false."*; calling `Conditional(P,Q).nl()` may return *"Provided that P, we know that Q."*. This template sampling, combined with optional adverb or negator insertion, gives LogicTree a high level of lexical diversity while preserving formal truth values.

## B   RELATED WORK

In this section, we review prior work related to our problem setting, including logical reasoning datasets (Section B.1) and reasoning-improvement methods (Section B.2).

### B.1   RELATED DATASETS

Logical reasoning datasets can broadly be categorized into three types. The first type focuses on deductive reasoning. The second type is based on grid-based logic puzzles. The third category comprises datasets based on multi-hop or strategic question answering. These datasets assess language models' logical capabilities from various perspectives, including formal logic, multi-step planning, structural induction, and strategy analysis. In addition, there are general-purpose reasoning datasets that are also frequently used to evaluate LLMs' logical reasoning abilities.

Table 10: Full No-CoT Prompt Template used for DRER training and evaluation.

---

**No-CoT Prompt**

**System Input**:
```
<|im_start|>system
You are a helpful assistant.  You answer questions by solely using
logical reasoning.
You will be given a paragraph of logical premises and a statement.
Perform logical reasoning strictly based on the premises using
propositional logic.
Assume all premises are true.  Do not rely on prior world
knowledge.

<answer> Fill in the final answer list for {num_q} question(s) here:
True, False or Unknown.  Like this:  [True, False...]  </answer>
You must choose one of the following answers:
- TRUE: if the premises entail the statement
- FALSE: if the premises contradict the statement
- UNKNOWN: if you cannot determine the truth value of the statement
based on the premises

<|im_end|>
```

**User Input**:
```
<|im_start|>user
Paragraph:  {paragraph}
{current_question}
<|im_end|>
<|im_start|>assistant
<answer>... </answer>
```

**Variable meanings**:
{num_q}: Number of questions in the current prompt.
{paragraph}: Paragraph containing the logical premises.
{current_question}: Statement whose truth value is to be evaluated.

---

### B.1.1 DEDUCTIVE REASONING

ConTRoL (Liu et al., 2021), consisting of 8,325 pairs of expert-designed datasets, is a challenging segment-level NLI dataset to evaluate model's contextual reasoning capacity from police recruitment tests. RuleTaker (Clark et al., 2020) is a benchmark dataset designed to test whether language models can logically reason about natural language rules and facts by determining whether the conclusions follow, do not follow, or are uncertain. LogiQA (Liu et al., 2020) is a benchmark of 8,678 civil service exam questions designed to evaluate models' reading comprehension and deductive reasoning across five logical types by requiring conclusion drawing from textual premises. LogiQA2.0 (Liu et al., 2023) is the enchanced version of LogiQA (Liu et al., 2020), featuring improved translations, expert-verified annotations, and new NLI tasks, designed to evaluate logical reasoning and reading comprehension in MRC and NLI formats. FOLIO Han et al. (2022) is an maually annotated dataset containing 1,430 logically complex natural language reasoning examples with first-order logic (FOL) annotations, designed to evaluate and benchmark the deductive reasoning and NL-FOL translation capabilities of Large Language models. PrOntoQA (Saparov & He, 2022) is a benchmark proposed in 2022 to evaluate LLMs' reasoning by generating question-answer pairs from first-order logic, revealing their struggles with multi-step proof planning despite valid individual steps. Compared with PrOntoQA (Saparov & He, 2022), PrOntoQA-OOD (Saparov et al., 2023) is designed to evaluate the general deductive reasoning abilities of LLMs by testing their ability to generalize to more complex, compositional proofs, particularly those that are out-of-distribution (OOD). JustLogic (Chen et al.,

---

**Algorithm 1 DRER: D**ynamic **R**easoning **E**fficiency **R**eward.

---

**Require:** Prompts $P = \{q_b\}_{b=1}^{B}$, ground-truth answers $Y^{\star} = \{a_b^{\star}\}_{b=1}^{B}$,
  1: policy $\pi_\theta$, rule reward $R_{\text{rule}}(\cdot)$, reasoning weight $\lambda_q$,
  2: bucket IDs $\{d_b\}_{b=1}^{B}$, bounds $\left(L_{\min}^{(d)}, L_{\max}^{(d)}\right)$, temperature $\tau$
**Ensure:** Advantages $A \in \mathbb{R}^{B \times L}$
    **(1) Build trajectories**
  3: $C \leftarrow \pi_\theta(P, \text{mode} = cot)$                                    $\triangleright$ CoT trajectories
  4: **for** $b = 1$ **to** $B$ **do**
  5:      $t_n[b] \leftarrow \text{NOCOTPROMPT}(q_b) \,\|\, \text{FORMATANSWER}(a_b^{\star})$
  6:      Replace answer span in $C[b]$ with $a_b^{\star} \rightarrow t_c[b]$; record span $\mathcal{A}_b$
  7: **end for**
    **(2) Reasoning-quality reward**
  8: **for** $b = 1$ **to** $B$ **do**
  9:      $\ell_c = \frac{1}{|\mathcal{A}_b|}\sum_{t\in\mathcal{A}_b} \log p_\theta(a_{b,t}^{\star} \mid t_c[b])$
 10:      $\ell_n = \frac{1}{|\mathcal{A}_b|}\sum_{t\in\mathcal{A}_b} \log p_\theta(a_{b,t}^{\star} \mid t_n[b])$
 11:      $R_q[b] \leftarrow \tanh(\ell_c - \ell_n)$
 12:      $R_{\text{seq}}[b] \leftarrow R_{\text{rule}}(C[b]) + \lambda_q R_q[b]$
 13:      Expand $R_{\text{seq}}[b]$ to token reward $r_{b,\cdot}$ on $C[b]$
 14: **end for**
    **(3) Group-wise normalisation**
 15: **for all** prompt group $g$ **do**
 16:      $\mu_g \leftarrow \text{mean}(r_{m,\cdot}),\ \sigma_g \leftarrow \text{std}(r_{m,\cdot})\quad (m\in g)$
 17:      **for** $m \in g$ **do**                                     $\triangleright$ raw advantage $\tilde{A}$
 18:          $\tilde{A}_{m,\cdot} \leftarrow \dfrac{r_{m,\cdot} - \mu_g}{\sigma_g + \varepsilon}$
 19:      **end for**
 20: **end for**
    **(4) Dynamic-length attenuation**
 21: **for** $b = 1$ **to** $B$ **do**
 22:      $\ell_b \leftarrow \text{LENGTH}(C[b]),\quad d \leftarrow d_b$
 23:      $g_b \leftarrow \exp\!\left(-\dfrac{\max\{0,\ L_{\min}^{(d)} - \ell_b,\ \ell_b - L_{\max}^{(d)}\}}{\tau}\right)$
 24:      $A_{b,\cdot} \leftarrow g_b \cdot \tilde{A}_{b,\cdot}$
 25: **end for**
 26: **return** $A$

---

2025) a generated deductive reasoning benchmark designed to evaluate LLMS, featuring high complexity, being independent of prior knowledge, and conducting in-depth error analysis in terms of reasoning depth and argumentative form.

However, the existing logical reasoning datasets still have some limitations. Most datasets have fixed or limited reasoning depth and breadth, which limits their ability to conduct a comprehensive evaluation of complex multi-step reasoning models. Many datasets entwine semantic information with logic, which may lead the model to rely on semantic cues rather than pure logical reasoning.

Furthermore, the majority focus only on final answer correctness, lacking assessment of the intermediate reasoning process and overall explanation quality.

In contrast, the LogicTree dataset we proposed has significant advantages: it is programmed and dynamically constructed, allowing for flexible control over the depth, breadth, and difficulty of inference; It separates semantics from logic to precisely evaluate pure deductive reasoning; It introduces a new logical consistency metric across multiple logical equivalence problems to measure the model's grasp of the underlying logical structure.

Table 11: Seven deductive paradigms that serve as the atomic reasoning units in LOGICTREE.

| Paradigm | Formal Schema | Surface Realisation |
|---|---|---|
| Modus Ponens | $(p \to q) \land p \Rightarrow q$ | If Alice studies, she will pass. Alice studies. Therefore, she will pass. |
| Modus Tollens | $(p \to q) \land \neg q \Rightarrow \neg p$ | If it rains, the road is wet. The road is not wet. Thus, it did not rain. |
| Hypothetical Syllogism | $(p \to q) \land (q \to r) \Rightarrow (p \to r)$ | If A wins, B celebrates. If B celebrates, C is happy. Hence, if A wins then C is happy. |
| Disjunctive Syllogism | $(p \lor q) \land \neg p \Rightarrow q$ | Either today is Monday or Tuesday. Today is not Monday. Therefore, today is Tuesday. |
| Reductio ad Absurdum | $(p \to q) \land (p \to \neg q) \Rightarrow \neg p$ | Assume the number is both even and odd. This leads to a contradiction. Thus, the number is not both even and odd. |
| Constructive Dilemma | $(p \to q) \land (r \to s) \land (p \lor r) \Rightarrow (q \lor s)$ | If it rains, we stay in; if it is sunny, we picnic. Either it rains or it is sunny. Hence, we either stay in or picnic. |
| Disjunction Elimination | $(p \lor q) \land (p \to s) \land (q \to s) \Rightarrow s$ | Either I study or I work. If I study, I will learn. If I work, I will learn. Thus, I will learn. |

Table 12: Primitive and compound proposition types used in LOGICTREE.

| Construct | Logical Form | Example Surface Realisation (EN) |
|---|---|---|
| Statement (atomic) | $p$ | *Alice studies.* |
| Negation | $\neg p$ | *It is **not** true that Alice studies.* |
| Conjunction | $P \land q$ | *Alice studies **and** Bob plays chess.* |
| Implication (Conditional) | $P \to q$ | *If it rains, **then** the road becomes wet.* |
| Inclusive Disjunction | $P \lor q$ | ***Either** today is Monday **or** Tuesday.* |

### B.1.2 GRID-BASED LOGIC PUZZLES

BoardgameQA(Kazemi et al., 2023) is a dataset designed to evaluate the reasoning ability of language models when dealing with contradictory information. GridPuzzle(Tyagi et al., 2024) is a dataset of grid-based logic puzzles designed to evaluate LLMs' structured, multi-step reasoning abilities through both final answers and detailed reasoning chains. The Knights and Knaves(Xie et al., 2025) dataset is an reasoning dataset designed to test logical deduction, where characters are either knights (truth-tellers) or knaves (liars), featuring controlled difficulty levels, procedural generation, and verifibility.

Existing datasets, such as GridPuzzle (Tyagi et al., 2024), Knights and Knaves (KK) (Xie et al.) provide valuable reasoning benchmarks, but they all have limitations. For example, KK (Xie et al.) entangles logical reasoning with semantic cues, taking the risk of rapid learning through keyword associations. Some logic puzzle focuses on the final answer without verifying the intermediate steps, allowing the model to guess without sufficient reasoning.

On the contrary, LogicTree evaluates the final and intermediate steps and executes the complete reasoning chain. It also introduces a logical consistency rate among variants of the same logical form and uses semantic-logical unentanglement to ensure that the model relies on reasoning rather than superficial clues.

### B.1.3 MULTI-HOP OR STRATEGIC QUESTION ANSWERING

HotpotQA (Yang et al., 2018) is a multi-hop question-answering dataset that requires reasoning across multiple documents and provides supporting facts to enhance the interpretability of the QA system. StrategyQA (Geva et al., 2021) is a benchmark dataset designed to evaluate implicit multi-step reasoning in LLMs across 15 domains and 13 strategies. SPAG (Cheng et al., 2024) is self-laying based adversarial language game dataset designed to enhance and evaluate the reasoning ability through a game involving indirect communication and strategic reasoning about hidden target words. LOGICGAME (Gui et al., 2024) is a benchmark designed to evaluate LLMs' ability to understand,

execute, and plan based on predefined rules through diverse, verifiable game scenarios requiring multi-step logical reasoning. AutoLogi (Zhu et al., 2025) is benchmark test for open-ended logic puzzles with controllable difficulty and program-based verification, designed to evaluate the reasoning ability of LLM.

Compared with datasets such as HotpotQA (Yang et al., 2018), StrategyQA (Geva et al., 2021), they emphasize various forms of multi-step or strategic reasoning across natural language problems, but there are still obvious limitations: The reasoning strategies in existing datasets are often broad and empirical rather than based on formal logical deduction frameworks (for example, StrategyQA (Geva et al., 2021) relies on heuristic and empirical categories). Many datasets focus on language pattern matching or cross-document evidence aggregation rather than verifying the true formal reasoning process (for example, HotpotQA (Yang et al., 2018)). LogicTree, on the other hand, strictly adheres to classical mathematical logic, adopting clear and well-defined deduction rules, and does not rely on common sense knowledge, providing a pure logical reasoning environment.

### B.1.4 General-Purpose Datasets

MMLU-Pro(Wang et al., 2024d) is an advanced benchmark of 12,000 expert-reviewed, 10-option questions across 14 disciplines, designed to better evaluate LLM performance with greater difficulty and reduced noise than the original MMLU (Hendrycks et al., 2021). However, it primarily evaluates broad knowledge and reasoning abilities rather than focusing on strong formal logical reasoning. Thus, it is not specifically designed to test models' capabilities in complex multi-step logical deduction.

## B.2 Related Reasoning Methods

Recent research has explored improving LLM reasoning through critique-based or reward-model-based mechanisms. Below we summarize the most relevant directions and clarify how our approach differs.

### B.2.1 Critique-Based Reasoning Approaches

Early self-improvement methods such as Self-Refine, Reflexion, and CRITIC require models to generate critique text to revise their own answers (Madaan et al., 2023; Shinn et al., 2023; Gou et al., 2023). Subsequent analyses report that such iterative critique loops can be unstable or rely on superficial linguistic artifacts rather than genuine logical reasoning (Huang et al., 2023; Valmeekam et al., 2023).

Other work focuses on supervised critique generation, such as Critique Fine-Tuning (CFT), which trains models to imitate human- or teacher-provided critique trajectories (Wang et al., 2025a;b). Similarly, Critique-Guided Distillation uses an external critic to score outputs and distills these scores into the model (Kapusuzoglu et al., 2025).

These methods rely on explicit critique traces or external critic models and supervise critique *content*. In contrast, DRER evaluates whether the reasoning chain itself improves the likelihood of the correct answer, without requiring critique generation or additional supervision.

### B.2.2 Positioning DRER Relative to CFT and CRL

Critique Reinforcement Learning (CRL) incorporates critiques into RL by rewarding models for predicting correct True/False judgments about candidate solutions (Ruan et al., 2025). CRL therefore optimizes judgment correctness, whereas DRER optimizes the causal contribution of reasoning steps via CoT–NoCoT likelihood margins.

CFT-based methods supervise the generation or imitation of critique traces (Wang et al., 2025a;b), while self-reflection methods rely on iterative critique production (Madaan et al., 2023; Shinn et al., 2023). DRER differs in that it introduces a counterfactual, gold-grounded reward that directly measures the usefulness of reasoning steps, without learning to critique or to judge solutions.

### B.2.3 PROCESS-LEVEL REINFORCEMENT LEARNING FOR REASONING

Another line of work improves reasoning through reinforcement learning that directly optimizes model behavior on reasoning tasks without relying on critique generation. Early RLHF-style approaches focus on outcome rewards (Schulman et al., 2017a) but do not supervise intermediate steps.

More recently, process-level RL methods such as GRPO and DAPO (Shao et al., 2024; Yu et al., 2025) use step-dependent rewards or decomposition strategies to encourage more stable reasoning trajectories. RLVR-style methods further incorporate structured or rule-based verification to provide process supervision (Wen et al., 2025). These approaches demonstrate that reinforcing intermediate reasoning behaviors can improve both accuracy and consistency.

DRER shares the goal of process-level supervision but differs fundamentally in how reasoning quality is evaluated: instead of using rule-based scoring or explicit correctness checks, DRER introduces a counterfactual, likelihood-based reward that measures whether the CoT reasoning trajectory increases model support for the correct answer. This avoids the need for handcrafted rules or verifiers while still providing a process-level training signal.

### B.2.4 REWARD-MODEL-BASED REASONING

Another family of methods trains reward models to evaluate reasoning steps or final answers (Wang et al., 2024c; Luo et al., 2024b). These systems can improve reasoning quality but require substantial labeled comparisons or step-by-step critiques. In contrast, DRER does not require a separate reward model; instead, it uses a counterfactual log-likelihood difference derived directly from the model's own outputs, providing a lighter-weight and verifiable training signal.

### B.2.5 OVERALL METHODOLOGICAL POSITIONING

Critique-based approaches supervise critique production or correctness, while reward-modeling approaches train external evaluators of reasoning quality. Process-oriented RL methods, such as RLVR-style training, supervise only the final answer.

DRER occupies a distinct space: it introduces a counterfactual, gold-grounded reward that measures whether the reasoning chain genuinely increases support for the correct answer. Thus, DRER complements rather than overlaps with critique-based or reward-modeling paradigms.

## C PROMPT TEMPLATES

Tables 9 and 10 list the exact prompts used in our experiments: a *Chain-of-Thought (CoT)* version that elicits step-by-step reasoning, and a *No-CoT* variant that asks for the final answer only. Curly-braced placeholders are replaced at runtime (`{paragraph}`, `{current_question}`, `{num_q}`). The two prompts share identical task instructions, so performance differences isolate the effect of showing or hiding the reasoning chain.

## D TRAINING DETAILS

### D.1 TRAINING SETTING

Table 13 records important training parameters. Experiments are conducted on $4 \times$H20 (80G) GPUs with CUDA 12.0, PyTorch 2.6.0, transformers 4.47.1. The Main Experiment phase (DAPO+DRER) trains for 400 training steps and takes approximately 50 hours. Training is carried out with a learning rate of $3 \times 10^{-7}$, a maximum response length of 4096 tokens, the batch size is 16 and 16 responses per prompt. For GRPO, the KL divergence coefficient is set to 0.001. In the DRER framework, we set $\lambda_q = 1$ and $\tau = 8$.

As shown in Equation 9, there are two parts of reward in DRER framework. Our Reasoning Quality Reward $R_q$ is range from $[-\lambda, \lambda]$ to measure whether those CoT tokens help to choose the correct answer. The general task reward $R_{task}$ depends on the specific training data, usually to verify the model's answer and format correctness. In our experiment, hyperparameter $\lambda$ is set to 1, the total task

Table 13: Important Training Parameters.

| Algorithm | Train Batch Size | Rollout N | KL Coef | Max Response Len |
|---|---|---|---|---|
| GRPO | 16 | 16 | 0.001 | 4096 |
| DAPO | 16 | 16 | – | 4096 |

reward is computed as:

$$R_{\text{task}} = S_{\text{format}} + S_{\text{answer}}$$

where the format score ($S_{\text{format}}$) evaluates whether the model's response adheres to the required output structure:

$$S_{\text{format}} = \begin{cases} 1, & \text{if format is correct} \\ -1, & \text{if format is incorrect} \end{cases}$$

And the answer score ($S_{\text{answer}}$) evaluates the correctness of the response content against the ground truth.

$$S_{\text{answer}} = \begin{cases} 2, & \text{if the answer fully matches the ground truth} \\ -1.5, & \text{if the answer partially mismatches the ground truth} \\ -2, & \text{if the answer cannot be parsed or is missing} \end{cases}$$

## D.2 TRAINING DYNAMICS

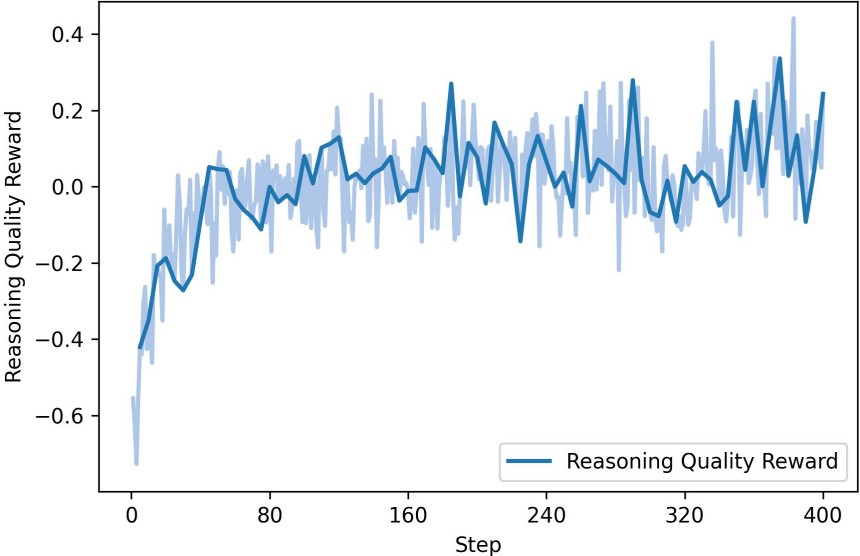

Figure 4: Reasoning quality reward on the LogicTree during post-training with DRER.

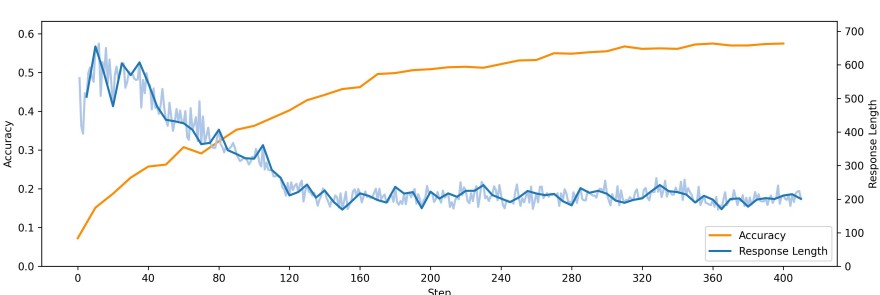

Figure 5: Training dynamic of the DAPO baseline with the DRER framework over 400 steps.

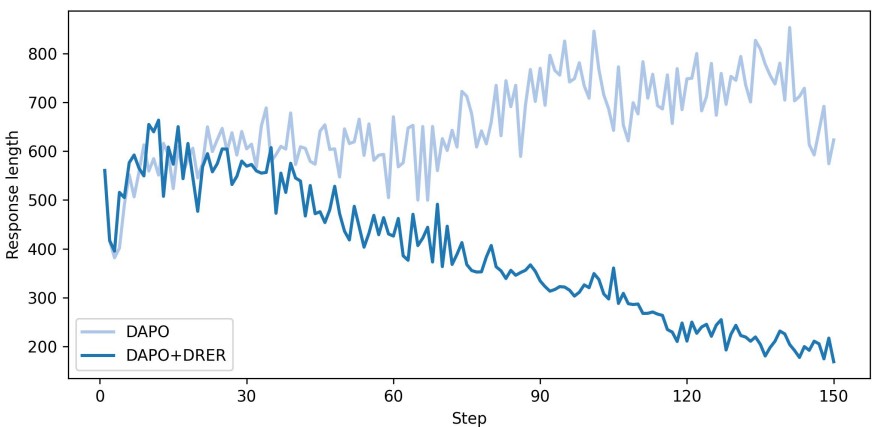

Figure 6: Comparison of response lengths over training steps between DAPO and DAPO+DRER. The integration of DRER leads to a reduction in response length, indicating enhanced efficiency with concise output.

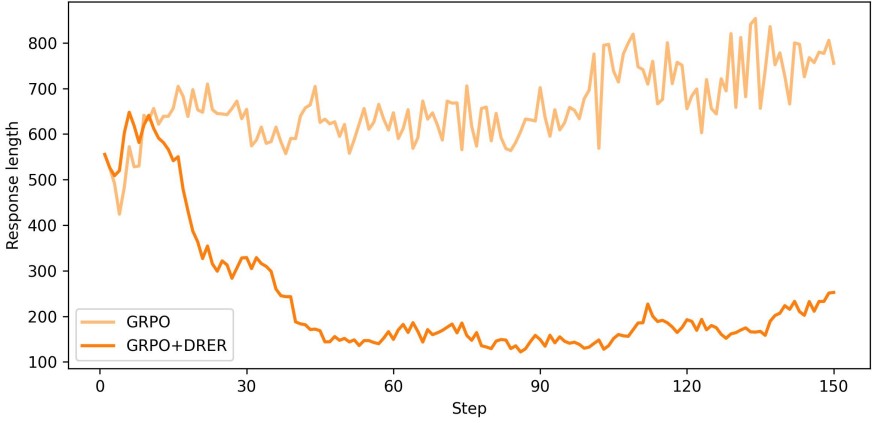

Figure 7: Comparison of response lengths over training steps between GRPO and GRPO+DRER. The integration of DRER leads to a reduction in response length, indicating enhanced efficiency with concise output.

# E SUPPLEMENTARY EXPERIMENTS

## E.1 EXPERIMENT A: GPT-5.1–BASED COT QUALITY SCORING

To further validate whether our Reward for Quality Reasoning (RQR) corresponds to genuine reasoning quality, we conduct an additional external evaluation using GPT-5.1.

We randomly sample 4000 chain-of-thought (CoT) trajectories from the evaluation set, including outputs from the base model, the DAPO-only model, and the DAPO+DRER model. GPT-5.1 is instructed to evaluate each trajectory in a step-wise manner. For every reasoning step, GPT-5.1 assigns binary judgments along three dimensions: *correctness*, *coherence*, and *necessity*. The detailed evaluation rubric and prompt are provided in Table 25, and a representative annotated example is shown in Table 24. For each trajectory, we aggregate the step-wise labels into a single CoT quality score, bucket the examples by this CoT Score, and compute the mean RQR within each bucket to examine how RQR correlates with externally assessed reasoning quality.

For a CoT consisting of $T$ steps, we define:

$$\text{Correctness} = \frac{1}{T}\sum_{t=1}^{T}\text{correctness}_t, \qquad \text{Necessity} = \frac{1}{T}\sum_{t=1}^{T}\text{necessity}_t.$$

To penalize chains whose logical flow breaks early, let

$$k = \min\{t \mid \text{coherence}_t = 0\}$$

be the index of the first coherence error. Coherence is defined as:

$$\text{Coherence} = \begin{cases} 1, & \text{if coherence}_t = 1 \ \forall t, \\ \alpha^{(T-k)}, & \text{otherwise,} \end{cases} \qquad \text{with } \alpha = 0.7.$$

We combine the three dimensions into a single CoT quality score:

$$\text{CoT Score} = 0.5 \cdot \text{Correctness} + 0.3 \cdot \text{Coherence} + 0.2 \cdot \text{Necessity}.$$

We bucket all examples by CoT Score and compute the mean RQR within each bucket.

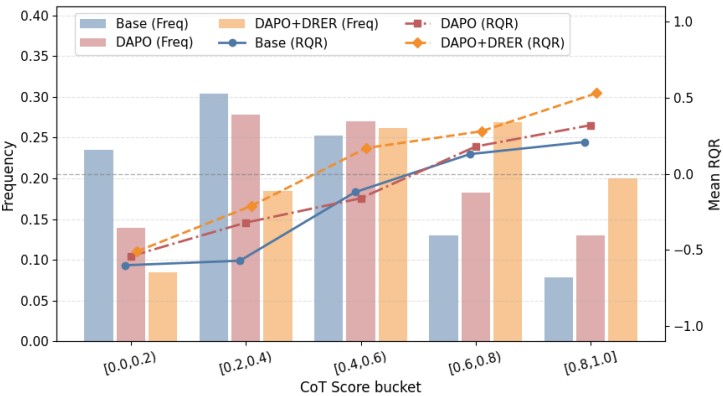

Figure 8: Comparison of CoT score distributions and corresponding mean RQR values across three training settings: Base model, DAPO-only, and DAPO+DRER. Bars represent the frequency of samples within each CoT score bucket, while the line plots show the mean RQR computed over the same buckets. DRER produces a clear shift toward higher-quality CoT trajectories and consistently higher RQR across all buckets.

The results reveal a clear and consistent trend: CoT trajectories with higher GPT-5.1 quality scores obtain substantially higher RQR values, whereas trajectories receiving low scores consistently yield lower RQR. After DRER training, both the distribution of GPT-5.1 CoT scores and the corresponding RQR values shift markedly toward higher-quality regions.

These observations indicate that RQR assigns larger rewards to more logically coherent and effective reasoning chains, demonstrating that the learned reward signal aligns with genuine reasoning quality rather than surface-level patterns.

Across both the DAPO-only and DAPO+DRER models, we observe that:

- Higher CoT Score consistently corresponds to higher RQR;

- DRER training increases RQR across all buckets, with the largest improvements in the high-quality CoT region.

Overall, these findings confirm that RQR is well aligned with GPT-5.1's step-wise evaluation of reasoning, capturing meaningful aspects of logical correctness and procedural validity.

### E.2 EXPERIMENT B: COT DISTURBANCE TEST

To assess whether the Reward for Quality Reasoning (RQR) is sensitive to the structural and semantic validity of reasoning trajectories, we conduct a controlled CoT–perturbation study on 4,000 randomly sampled questions from our evaluation set.

For each question, we construct three variants of the chain-of-thought (CoT):

- **Original CoT**: the unmodified reasoning trajectory generated by the model.

- **Shuffled CoT**: a sentence-level random permutation of the same trajectory, disrupting logical order while preserving content.

- **Cross-question CoT**: a CoT drawn from a different evaluation question, approximately length-matched but semantically unrelated.

For each variant, we compute the RQR defined in Eq. (8). Table 14 reports the mean RQR, standard deviation, and proportion of positive RQR values.

| CoT Variant | Mean RQR ↑ | Std RQR | % RQR $> 0$ ↑ |
|---|---|---|---|
| Original CoT | 0.29 | 0.42 | 73% |
| Shuffled CoT | 0.08 | 0.31 | 41% |
| Cross-question CoT | -0.34 | 0.33 | 7% |

Table 14: Experiment B: RQR under different CoT perturbations on 2,000 randomly sampled evaluation questions. The ordering Original > Shuffled > Cross demonstrates that RQR aligns with reasoning quality and task relevance.

**Summary of Results.** These results indicate that RQR exhibits clear sensitivity to both the semantic relevance and structural coherence of the reasoning chain, rather than displaying a simple preference for the presence of CoT tokens. The significant differences across perturbation types suggest that RQR captures the degree to which intermediate reasoning steps either support or hinder the correct answer probability, reflecting their contribution in the problem-solving process.

### E.3 EXPERIMENT C: ANALYSIS OF THE EFFECTS OF COT

Table 15: Average $\ell_{\text{CoT}} - \ell_{\text{NoCoT}}$ by answer transition in GSM8K.

| Original ↓ / With CoT → | Wrong (W) | Correct (R) |
|---|---|---|
| Wrong (W) | -4.32 | 2.46 |
| Correct (R) | -5.00 | -0.47 |

Table 16: Average $\ell_{\text{CoT}} - \ell_{\text{NoCoT}}$ by answer transition in LogicTree.

| Original ↓ / With CoT → | Wrong (W) | Correct (R) |
|---|---|---|
| Wrong (W) | -1.13 | 1.81 |
| Correct (R) | -3.79 | -4.76 |

Table 17: Answer-transition proportions conditioned on the sign of $\Delta\ell = \ell_{\text{CoT}} - \ell_{\text{NoCoT}}$ on GSM8K (N=500). $p(\text{W}\rightarrow\text{R})$ is the fix rate; $p(\text{R}\rightarrow\text{W})$ is the break rate.

| Group by $\Delta\ell$ sign | #Instances | Mean $\Delta\ell$ | $p(\text{W}\rightarrow\text{R})$ | $p(\text{R}\rightarrow\text{W})$ |
|---|---|---|---|---|
| $\Delta\ell > 0$ (CoT favored) | 140 | +2.20 | 0.74 | 0.02 |
| $\Delta\ell < 0$ (NoCoT favored) | 360 | -2.60 | 0.02 | 0.24 |

### E.4 EXPERIMENT D: LOGICTREE EVALUATION

This section lists full evaluation on LogicTree as logic benchmark.

Table 19 exhibits the full evaluation data of Accuracy on LogicTree benchmark across various reasoning depths.

Table 20 presents the complete evaluation data of Consistency Ratio on LogicTree benchmark.

Figure 9 plots the complete evaluation data of *Fβ-Score*, which provides a balanced metric to compare the comprehensive performance across those LLMs.

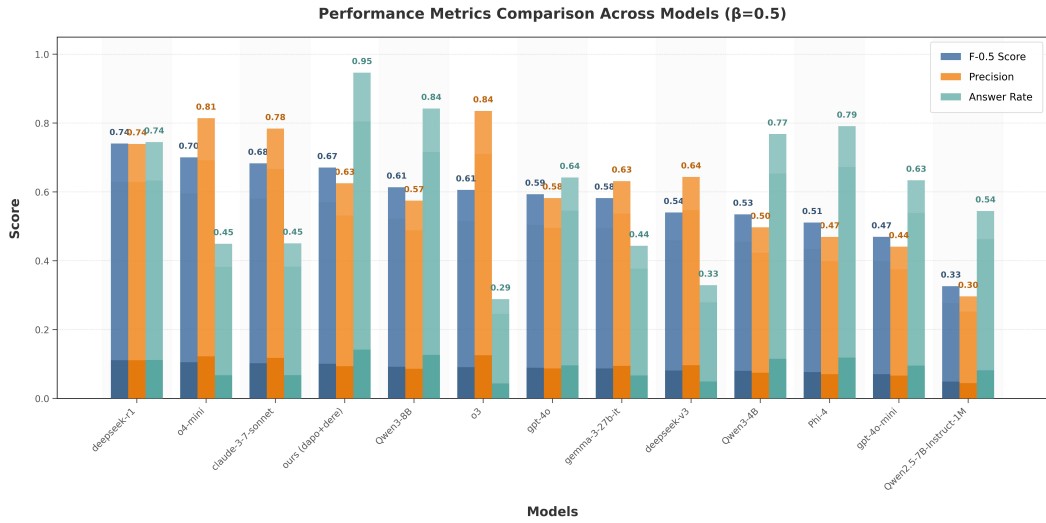

Figure 9: *Fβ-Score*, Answer Rate and Precision metrics Comparison across various models.

Figure 10 shows the distribution of those deduction logical key words in LLMs response.

Figure 11 compares the reasoning token efficiency between DeepSeek-R1 and our model.

Figure 12 compares the output distribution between models trained with DAPO and DAPO+DRER respectively. The DAPO+DRER model demonstrates significantly higher confidence in correct answers, as shown by a strong concentration of predictions on the fully correct label set ([true, true, true]). In contrast, the baseline DAPO model produces more scattered outputs, indicating lower certainty. This highlights the effectiveness of DRER in combination with CoT reasoning for improving answer consistency and correctness.

Figure 12 compares the output distribution between base model and variant trained with DAPO+DRER. The DAPO+DRER model produces highly concentrated predictions on the fully correct label ([true, true, true]), indicating strong confidence and consistency. In contrast, Qwen2.5-

Table 18: Answer-transition proportions conditioned on the sign of $\Delta\ell = \ell_{\text{CoT}} - \ell_{\text{NoCoT}}$ on LogicTree (N=500). $p(\text{W}\rightarrow\text{R})$ is the fix rate; $p(\text{R}\rightarrow\text{W})$ is the break rate.

| Group by $\Delta\ell$ sign | #Instances | Mean $\Delta\ell$ | $p(\text{W}\rightarrow\text{R})$ | $p(\text{R}\rightarrow\text{W})$ |
|---|---|---|---|---|
| $\Delta\ell > 0$ (CoT favored) | 120 | +1.70 | 0.67 | 0.04 |
| $\Delta\ell < 0$ (NoCoT favored) | 380 | -3.90 | 0.05 | 0.23 |

Table 19: Comparison of LRM's(above) and LLM's(below) accuracy on LogicTree across various logical depth.

| Model | 1 | 2 | 3 | 4 | 5 | 6 | 7 | 8 | Avg. |
|---|---|---|---|---|---|---|---|---|---|
| Qwen3-235B-A22B | **0.96** | **0.83** | 0.66 | **0.71** | 0.46 | 0.32 | 0.25 | 0.07 | 0.53 |
| Deepseek-R1 | 0.85 | 0.76 | 0.61 | 0.47 | 0.36 | 0.18 | 0.19 | 0.07 | 0.44 |
| Claude-3.7-Sonnet | 0.76 | 0.67 | 0.21 | 0.10 | 0.07 | 0.02 | 0.02 | 0.00 | 0.23 |
| Qwen3-8B | 0.86 | 0.83 | 0.49 | 0.44 | 0.32 | 0.11 | 0.14 | 0.08 | 0.41 |
| GPT-o4-mini | 0.74 | 0.64 | 0.25 | 0.20 | 0.10 | 0.06 | 0.05 | 0.02 | 0.26 |
| GPT-o3-mini | 0.66 | 0.56 | 0.07 | 0.07 | 0.03 | 0.02 | 0.01 | 0.00 | 0.18 |
| Qwen3-4B | 0.74 | 0.74 | 0.39 | 0.29 | 0.29 | 0.06 | 0.09 | 0.04 | 0.33 |
| Gemini-2.5-Flash-Preview | 0.86 | 0.64 | 0.41 | 0.31 | 0.24 | 0.11 | 0.06 | 0.00 | 0.33 |
| GPT-4o | 0.63 | 0.60 | 0.28 | 0.13 | 0.13 | 0.00 | 0.00 | 0.00 | 0.22 |
| Phi-4-14B | 0.72 | 0.67 | 0.31 | 0.27 | 0.19 | 0.04 | 0.01 | 0.01 | 0.28 |
| Gemma-3-27B | 0.65 | 0.41 | 0.15 | 0.04 | 0.00 | 0.00 | 0.00 | 0.00 | 0.16 |
| Deepseek-v3 | 0.39 | 0.24 | 0.05 | 0.06 | 0.00 | 0.00 | 0.00 | 0.00 | 0.09 |
| GPT-4o-mini | 0.44 | 0.24 | 0.27 | 0.11 | 0.12 | 0.02 | 0.02 | 0.01 | 0.15 |
| Qwen2.5-7B-Instruct-1M | 0.36 | 0.29 | 0.15 | 0.12 | 0.08 | 0.01 | 0.01 | 0.00 | 0.13 |
| GRPO | 0.81 | 0.71 | 0.58 | 0.42 | 0.45 | 0.20 | 0.20 | 0.11 | 0.45 |
| GRPO+DRER | 0.87 | 0.75 | 0.69 | 0.54 | 0.61 | 0.35 | 0.27 | 0.22 | 0.54 |
| DAPO | 0.88 | 0.73 | 0.66 | 0.47 | 0.60 | 0.36 | 0.23 | 0.20 | 0.52 |
| **DAPO+DRER (Ours)** | 0.90 | **0.83** | **0.76** | 0.59 | **0.67** | **0.45** | **0.31** | **0.31** | **0.60**$^{\uparrow 0.47}$ |

7B-Instruct-1M predictions are widely dispersed across incorrect and partially correct categories, reflecting lower answer certainty. This highlights the effectiveness of DRER combined with CoT in guiding the model toward accurate and confident output.

Tables 21 records the average evaluation results on 15 graduate students who had received systematic training in mathematical logic or introductory logic courses. The results show that for problems of simple to moderate difficulty (reasoning depth 1–5), human participants consistently identified the implicit logical rules and produced correct answers. For deeper reasoning levels (6–8), although the problems remain theoretically solvable, the context length can exceed 1k tokens, making manual step-by-step deduction extremely tedious and error-prone. For this reason, depth-6–8 questions were excluded from human testing.

Tables 22 shows the exactly models' name and snapshot that we evaluated in experiment.

# F    LIMITATIONS

Despite the empirical gains achieved by DRER and LogicTree, several limitations remain:

- **Logic coverage.** LogicTree is limited to the deductive reasoning paradigm, while more diverse forms such as analogical reasoning, inductive reasoning, or traceable reasoning have not yet been evaluated.
- **Model scale and cost.** All experiments use **Qwen-2.5-7B-Instruct-1M** as backbone. The memory and latency overhead of token-level rewards on 70 B-scale or MoE models is unknown and may be prohibitive.
- **Evaluation bias.** Training and evaluation rely on an automatic logic verifier and confidence scores; no human preference or chain-quality annotation is included, which may overlook subjective aspects of reasoning quality.

Table 20: Comparison of Consistency Ratio on LogicTree across various logical depth.

| Model | 1 | 2 | 3 | 4 | 5 | 6 | 7 | 8 | Avg. |
|---|---|---|---|---|---|---|---|---|---|
| Qwen3-235B-A22B | **0.90** | 0.65 | 0.30 | **0.50** | 0.15 | 0.00 | 0.05 | 0.00 | 0.32 |
| Deepseek-R1 | 0.70 | 0.55 | 0.20 | 0.15 | 0.10 | 0.00 | 0.05 | 0.00 | 0.22 |
| Claude-3.7-Sonnet | 0.65 | 0.35 | 0.00 | 0.00 | 0.00 | 0.00 | 0.00 | 0.00 | 0.12 |
| Qwen3-8B | 0.65 | **0.70** | 0.05 | 0.05 | 0.05 | 0.00 | 0.00 | 0.00 | 0.19 |
| GPT-o4-mini | 0.50 | 0.35 | 0.00 | 0.05 | 0.00 | 0.00 | 0.00 | 0.00 | 0.11 |
| GPT-o3-mini | 0.45 | 0.30 | 0.00 | 0.00 | 0.00 | 0.00 | 0.00 | 0.00 | 0.09 |
| Qwen3-4B | 0.40 | 0.30 | 0.05 | 0.05 | 0.00 | 0.00 | 0.00 | 0.00 | 0.10 |
| Gemini-2.5-Flash-Preview | 0.75 | 0.50 | 0.15 | 0.05 | 0.00 | 0.00 | 0.00 | 0.00 | 0.18 |
| GPT-4o | 0.40 | 0.35 | 0.00 | 0.05 | 0.00 | 0.00 | 0.00 | 0.00 | 0.10 |
| Phi-4-14 | 0.35 | 0.35 | 0.05 | 0.05 | 0.00 | 0.00 | 0.00 | 0.00 | 0.10 |
| Gemma-3-27B | 0.25 | 0.20 | 0.00 | 0.00 | 0.00 | 0.00 | 0.00 | 0.00 | 0.06 |
| Deepseek-v3 | 0.15 | 0.00 | 0.00 | 0.00 | 0.00 | 0.00 | 0.00 | 0.00 | 0.02 |
| GPT-4o-mini | 0.10 | 0.10 | 0.00 | 0.00 | 0.00 | 0.00 | 0.00 | 0.00 | 0.03 |
| Qwen2.5-7B-Instruct-1M | 0.05 | 0.00 | 0.00 | 0.00 | 0.00 | 0.00 | 0.00 | 0.00 | 0.01 |
| GRPO | 0.55 | 0.50 | 0.40 | 0.25 | 0.45 | 0.20 | **0.15** | 0.00 | 0.29 |
| GRPO+DRER | 0.65 | 0.50 | 0.25 | 0.25 | 0.25 | 0.10 | 0.00 | 0.00 | 0.25 |
| DAPO | 0.65 | 0.45 | 0.45 | 0.20 | **0.50** | 0.10 | 0.05 | **0.10** | 0.31 |
| **DAPO+DRER (Ours)** | 0.70 | **0.70** | **0.60** | 0.35 | **0.50** | **0.35** | 0.00 | **0.10** | $0.41^{\uparrow 0.4}$ |

Table 21: Comparison of LLM and Human accuracy on LogicTree across various logical depth.

| Model | 1 | 2 | 3 | 4 | 5 | 6 | 7 | 8 | Avg. |
|---|---|---|---|---|---|---|---|---|---|
| Qwen3-235B-A22B | 0.96 | 0.83 | 0.66 | 0.71 | 0.46 | 0.32 | 0.25 | 0.07 | 0.53 |
| Deepseek-R1 | 0.85 | 0.76 | 0.61 | 0.47 | 0.36 | 0.18 | 0.19 | 0.07 | 0.44 |
| Claude-3.7-Sonnet | 0.76 | 0.67 | 0.21 | 0.10 | 0.07 | 0.02 | 0.02 | 0.00 | 0.23 |
| GPT-o4-mini | 0.74 | 0.64 | 0.25 | 0.20 | 0.10 | 0.06 | 0.05 | 0.02 | 0.26 |
| DAPO+DRER (Ours) | 0.90 | 0.83 | 0.76 | 0.59 | 0.67 | 0.45 | 0.31 | 0.31 | 0.60 |
| Human | 1.00 | 1.00 | 0.98 | 0.93 | 0.85 | - | - | - | - |

Table 22: Details of the organization and model source (model version for proprietary models, and Huggingface model name for open-source models) for the LLMs evaluated in LogicTree.

| Model | Organization | Size | Notes | Source |
|---|---|---|---|---|
| DeepSeek-R1 | DeepSeek | 671B | MoE | deepseek-ai/DeepSeek-R1 |
| DeepSeek-V3 | DeepSeek | 671B | MoE | deepseek-ai/DeepSeek-V3 |
| Claude 3.7 Sonnet | Anthropic | – | | claude-3-7-sonnet-20250219 |
| Gemini 2.0 Flash Thinking Preview | Google | – | | gemini-2.5-flash-preview-04-17 |
| Gemma-3-27B | Google | 27B | | google/gemma-3-27b-it |
| Qwen3-235B-A22B | Alibaba | 235B | MoE | qwen3-235b-a22b |
| Qwen3-30B-A3B | Alibaba | 30B | MoE | qwen3-30b-a3b |
| Qwe3-8B | Alibaba | – | | qwen3-8b |
| Qwen3-4B | Alibaba | – | | qwen3-4b |
| Qwen2.5-7B-Instruct-1M | Alibaba | – | MoE | qwen2.5-7b-instruct-1m |
| Phi-4-14B | Microsoft | 14B | | microsoft/phi-4 |
| GPT-o4-mini | OpenAI | – | | o4-mini-2025-04-16 |
| GPT-o3 | OpenAI | – | | o3-mini-2025-01-31 |
| GPT-4o-mini | OpenAI | – | | gpt-4o-mini-2024-07-18 |
| GPT-4o | OpenAI | – | | gpt-4o-2024-11-20 |

- **Synthetic corpus and social bias.** LogicTree sentences are synthetically generated; potential social biases or misuse risks in real-world deployments have not been systematically analysed.

In future work we plan to extend DRER to higher-order logic, explore low-cost reward approximations, and incorporate human evaluation and bias auditing to mitigate these limitations.

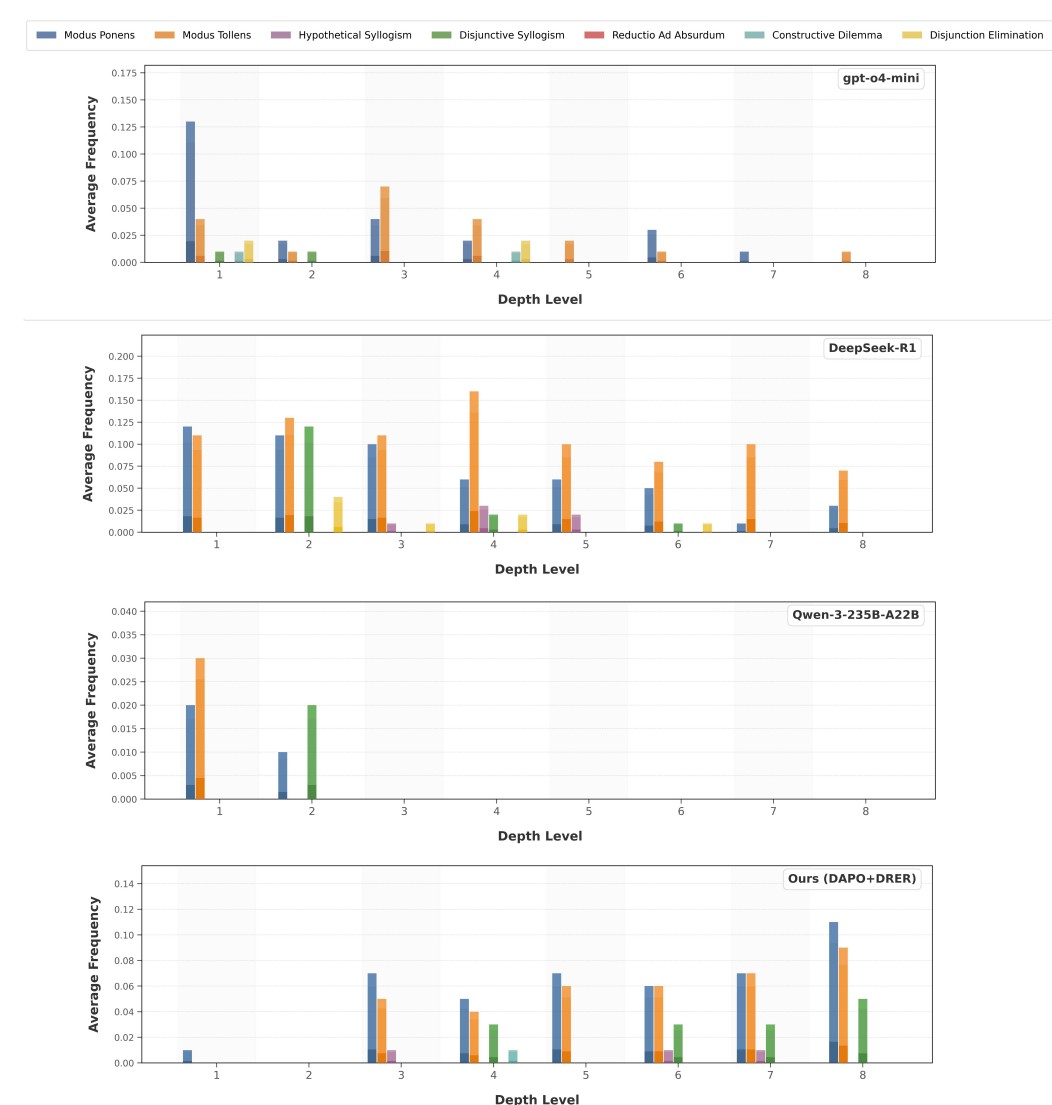

Figure 10: Word frequencies of seven deductive reasoning terms explicitly mentioned in LLMs response DRER.

# G  BROADER IMPACT

Our work aims to align large language models with formal logical principles, potentially improving the reliability and interpretability of machine reasoning. By releasing the LOGICTREE dataset and DRER code under a permissive licence, we enable researchers and practitioners to build verifiable agents for education, scientific discovery, and safety- critical auditing, where transparent deductive chains are preferable to opaque heuristics.

## G.1  POSITIVE SOCIETAL OUTCOMES.

A reasoning-aligned model can serve as a didactic tutor in introductory logic courses, assist engineers in detecting faulty assumptions in software specifications, and support legal or medical professionals by highlighting which premises lead to a conclusion rather than merely producing an answer. The synthetic nature of LOGICTREE limits exposure to personal data and reduces the risk of privacy leaks.

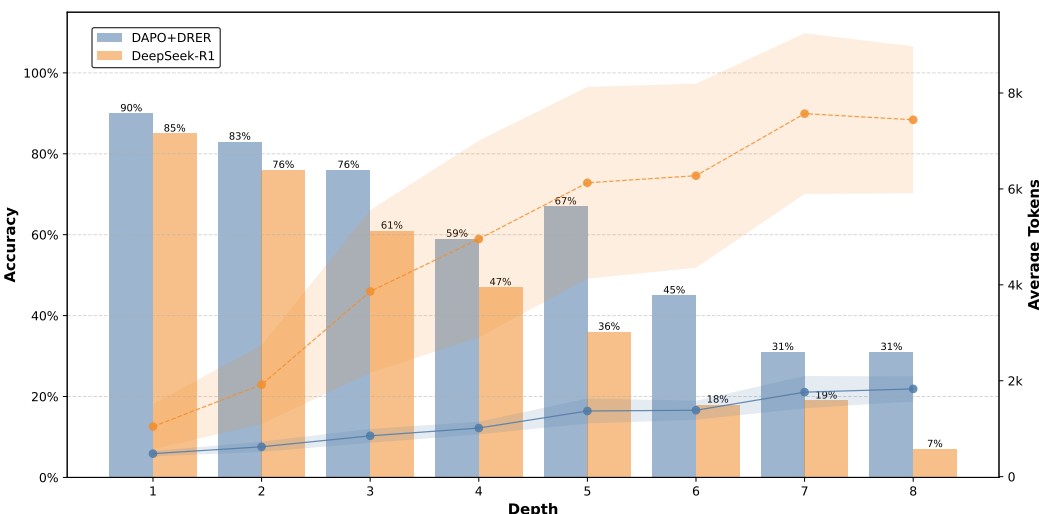

Figure 11: Comparison of DeepSeek-R1's and our model's accuracy and average response token on LogicTree.

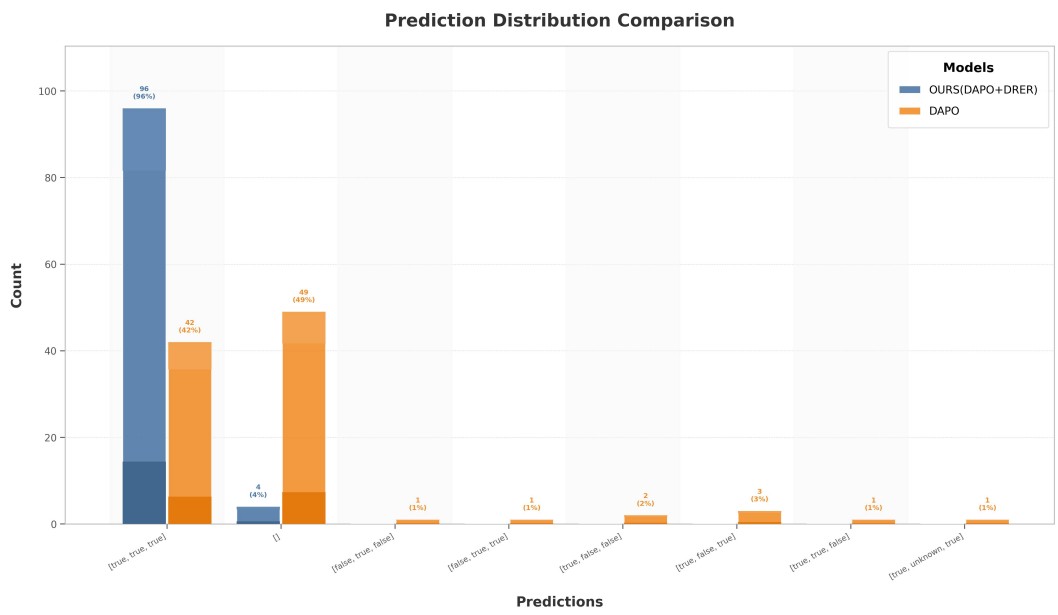

Figure 12: Prediction distribution comparison between DAPO and DAPO+DRER under Chain-of-Thought (CoT) prompting.

## G.2 POTENTIAL RISKS.

More persuasive and logically consistent outputs could be weaponised for misinformation or overly authoritative automation. Over-reliance on synthetic benchmarks might also hide biases that appear in real-world discourse. Furthermore, token-level reward signals expose fine-grained model behaviour, which could be exploited to reverse-engineer proprietary system prompts.

## G.3 MITIGATIONS.

We distribute our resources with an explicit no-malicious-use clause, encourage downstream users to apply bias and misinformation audits, and recommend human oversight for high-stakes deployment.

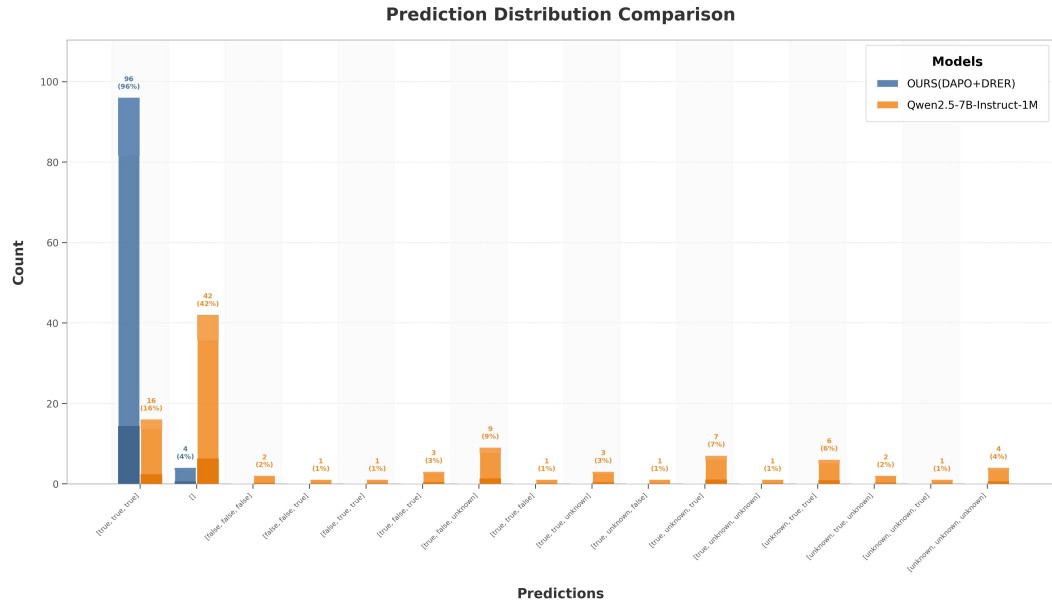

Figure 13: Prediction distribution comparison between our model (DAPO+DRER) and Qwen2.5-7B-Instruct-1M under Chain-of-Thought (CoT) prompting.

Future work will extend DRER to real-world corpora and incorporate human preference feedback, allowing broader yet safer adoption of reasoning-aligned reinforcement learning.

## H CASE STUDY: CHAIN-OF-THOUGHT QUALITY

To complement the quantitative evaluation, we present a case study based on the example shown in Table 23, which compares reasoning traces produced by different models on the same LogicTree instance. This case study highlights how DRER improves both the clarity and efficiency of chain-of-thought (CoT) reasoning.

### H.1 DAPO+DRER (OURS)

Driven by DRER's *token-level reward* and *dynamic-length mask*, the trace is both concise and transparent: six orderly steps map directly onto the formal pipeline *identify rule → resolve disjunction → propagate truth*. Expressions such as "apply transitivity" signal an explicit shift toward symbolic reasoning, the behaviour DRER is designed to promote. At roughly ∼70 tokens—far shorter than the 100+ tokens typical of vanilla DAPO—the chain remains fully verifiable, demonstrating DRER's combined gains in *effectiveness* and *efficiency*.

### H.2 O4MINI

Although it yields the correct answer, steps 2–4 compress several entailments into a single sentence and omit rule names, reducing transparency. With a length of about ∼40 tokens, it illustrates the "shallow-but-correct" pattern observed in §4.2.

### H.3 QWEN2.5-7B-INSTRUCT

This trace shows the greatest *length drift*: more than 110 tokens, many of them descriptive filler unrelated to logic, matching the "decorative-token inflation" failure mode in our diagnostics. Despite some correct premise restatement, the model ultimately outputs `Unknown`, confirming that verbosity does not equal confidence.

Table 23: Four-model CoT Comparison (Prompt-style)

---

**Four-model CoT Comparison**

**Question**
It is a common misconception that if *ribbons relate post : How to sew trims*, then *titanium dioxide and zinc oxide also functioned in this fashion.*

**dapo+drer**
*Chain-of-Thought*

1. Analyze the logical structure of the premises.
2. Identify conditional statements and their parts.
3. Resolve the disjunction via the invertebrate–coral premise.
4. Propagate truth to the olive-oil statement.
5. Apply transitivity to obtain *ribbons* $\rightarrow$ TiO$_2$/ZnO.
6. Compare with the question's claim.

**Answer**: **False**

---

**Qwen2.5-7B-Instruct-1M**
*Chain-of-Thought*

Let's analyze step by step: 1) Coral reefs need light and thus occur only in shallow waters. 2) Olive oil is heat-stable. 3) If ribbons $\rightarrow$ TiO$_2$/ZnO. 4) "Anus on top" is false $\Rightarrow$ coral-reef statement true. 5) No evidence of misconception. Hence information is insufficient.

**Answer**: **Unknown**

---

**o4mini**
*Chain-of-Thought*

1. "Anus on top" is false.
2. Therefore coral-reef-light is true.
3. Thus olive-oil statement is true.
4. Hence conditional *ribbons* $\rightarrow$ TiO$_2$/ZnO is true.
5. The question claims this conditional is a misconception.
6. Contradiction $\Rightarrow$ question is false.

**Answer**: **False**

---

**deepseekr1**
*Chain-of-Thought*

I need to analyze consistency; I will break down each conditional and test with the premises.

**Answer**: **False**

---

## H.4 DEEPSEEK-R1

Provides only a meta-level remark ("break down each conditional") before jumping to the answer, leaving the derivation invisible; such hidden reasoning yields the lowest Consistency Ratio in our evaluation across paraphrastic variants.

## H.5 SUMMARY.

**dapo+drer** offers the clearest, rule-grounded, and length-controlled chain of thought; **o4mini** is concise but omits warrants and lacks length regularisation; **Qwen2.5-7B-Instruct** is verbose yet uncertain; and **DeepSeek-R1** lacks an explicit chain. The contrast underscores DRER's targeted improvements in symbolic clarity, response economy, and process–outcome alignment.

Table 24: CoT Comparison (DAPO and DAPO+DRER)

---

**CoT on AIME (DAPO)**

**Question**
Let $x$, $y$, and $z$ be positive real numbers that satisfy the following system of equations:

$$\log_2\left(\frac{x}{yz}\right) = \frac{1}{2}$$

$$\log_2\left(\frac{y}{xz}\right) = \frac{1}{3}$$

$$\log_2\left(\frac{z}{xy}\right) = \frac{1}{4}$$

Then the value of $\left|\log_2\left(x^4 y^3 z^2\right)\right|$ is $\frac{m}{n}$, where $m$ and $n$ are relatively prime positive integers. Find $m + n$.

---

**DAPO**
Given the system of equations:

$$\log_2\left(\frac{x}{yz}\right) = \frac{1}{2}, \qquad \log_2\left(\frac{y}{xz}\right) = \frac{1}{3}, \qquad \log_2\left(\frac{z}{xy}\right) = \frac{1}{4},$$

we rewrite each in exponential form:

$$\frac{x}{yz} = 2^{1/2}, \quad \frac{y}{xz} = 2^{1/3}, \quad \frac{z}{xy} = 2^{1/4}.$$

Multiplying all three equations:

$$\left(\frac{x}{yz}\right)\left(\frac{y}{xz}\right)\left(\frac{z}{xy}\right) = 2^{1/2+1/3+1/4},$$

and the left-hand side simplifies to

$$\frac{xyz}{x^2 y^2 z^2} = \frac{1}{xyz},$$

so

$$\frac{1}{xyz} = 2^{13/12} \quad \Rightarrow \quad xyz = 2^{-13/12}.$$

Let $\log_2 x = a$, $\log_2 y = b$, $\log_2 z = c$. Then

$$a + b + c = \log_2(xyz) = -\frac{13}{12}.$$

Using the original equations:

$$a - (b + c) = \frac{1}{2}, \qquad b - (a + c) = \frac{1}{3}, \qquad c - (a + b) = \frac{1}{4}.$$

However, the incorrect solution attempted to compute:

$$\textcolor{red}{4a + 3b + 2c = 4\left(\frac{1}{2}\right) + 3\left(\frac{1}{3}\right) + 2\left(\frac{1}{4}\right)} \qquad \textcolor{red}{\text{Unsupported Equation}}$$

$$\textcolor{red}{= 2 + 1 + \frac{1}{2} = \frac{7}{2},}$$

which assumes

$$\textcolor{red}{a = \frac{1}{2}, \quad b = \frac{1}{3}, \quad c = \frac{1}{4}}$$

Table 24: CoT Comparison (DAPO and DAPO+DRER) (continued)

---

**CoT on AIME (DAPO)**

Thus the final claim:
$$\left|\log_2(x^4y^3z^2)\right| = \frac{7}{2}$$
$$m + n = 9$$

**Answer**: **9** ($\times$)

---

**DAPO+DRER**
To solve the given system of equations involving logarithms, we start by letting $a = \log_2 x$, $b = \log_2 y$, and $c = \log_2 z$.
The system becomes:
$$a - b - c = \frac{1}{2}, \qquad b - a - c = \frac{1}{3}, \qquad c - a - b = \frac{1}{4}.$$

Solving:
$$(a - b - c) + (b - a - c) = \frac{1}{2} + \frac{1}{3} \implies -2c = \frac{5}{6} \implies c = -\frac{5}{12}.$$

Substitute into the third equation:
$$-\frac{5}{12} - a - b = \frac{1}{4} \implies a + b = -\frac{2}{3}.$$

Substitute into the first equation:
$$a - b + \frac{5}{12} = \frac{1}{2} \implies a - b = \frac{1}{12}.$$

Solving the two-equation system:
$$a = -\frac{7}{24}, \qquad b = -\frac{3}{8}.$$

Thus:
$$c = -\frac{5}{12}.$$

Compute:
$$\log_2(x^4y^3z^2) = 4a + 3b + 2c = -\frac{25}{8}.$$

Since $\frac{m}{n} = \frac{25}{8}$, we have $m + n = 25 + 8 = 33$.

**Answer**: **33** ($\checkmark$)

Table 25: GPT-5.1 prompt for scoring CoT trajectories

---

**GPT-5.1 Prompt**

You are an expert judge for mathematical and logical reasoning steps.
You will be given:

- A problem statement (QUESTION).
- A full sequence of reasoning steps.
- The current reasoning step to evaluate (CURRENT_STEP), which is step {t}.

1. **correctness**: whether the content inside this step is mathematically or logically correct.
2. **coherence**: whether this step is consistent with the QUESTION and ALL_STEPS, and a reasonable next move.
3. **necessity**: whether this step contributes essential progress toward solving the problem.

**Strict Scoring Rubric [correctness]**

Score 1 if:

- There is no mathematical or logical error in this step, and
- It does not contradict the QUESTION or earlier correct steps.

Score 0 if:

- There is an algebraic or logical mistake, or
- A rule is misapplied, or
- The step contradicts the problem statement or prior correct steps.

**[coherence]**
Score 1 if:

- The step naturally follows from the QUESTION and ALL_STEPS, and
- Any new notation or assumptions are properly introduced.

Score 0 if:

- The step makes an unjustified assumption or conclusion, or
- It reverses earlier conclusions without reason, or
- It is disconnected from the reasoning flow.

**[necessity]**
Score 1 if:

- The step introduces new, nontrivial information or structure used later, or
- Removing the step would make the solution less complete or harder to follow.

Score 0 if:

- The step merely restates previous information, or
- It is meta-commentary, or
- It explores a direction not used in the main reasoning.

---

**GPT-5.1 Prompt**

**Output Format (Mandatory).**

You must output exactly one JSON dictionary with the following four fields:

```
{
  "correctness": 0/1,
  "coherence": 0/1,
  "necessity": 0/1,
  "analysis": "2--4 sentences explaining your scores."
}
```

Rules:

- Output must be valid JSON.
- Only these four fields may appear.
- No lists, markdown, backticks, or extra commentary.

**Final Instruction.**

Evaluate the current step:

```
QUESTION: {QUESTION}

ALL_STEPS: {ALL_STEPS}

CURRENT_STEP (step {t}): {CURRENT_STEP}
```

# I LLM USAGE

Large Language Models (LLMs) were used to aid in the writing and polishing of the manuscript. Specifically, we used an LLM to assist in refining the language, improving readability, and ensuring clarity in various sections of the paper. The model helped with tasks such as sentence rephrasing, grammar checking, and enhancing the overall flow of the text.

It is important to note that the LLM was not involved in the ideation, research methodology, or experimental design. All research concepts, ideas, and analyses were developed and conducted by the authors. The contributions of the LLM were solely focused on improving the linguistic quality of the paper, with no involvement in the scientific content or data analysis.

The authors take full responsibility for the content of the manuscript, including any text generated or polished by the LLM. We have ensured that the LLM-generated text adheres to ethical guidelines and does not contribute to plagiarism or scientific misconduct.

