# OpenReview forum: "Rethinking Reasoning Quality in Large Language Models through Enhanced Chain-of-Thought via RL"
_ICLR.cc/2026/Conference — Submitted to ICLR 2026_

### Official Review · Reviewer_Vyr9 · 2025-10-31

**Soundness:** 3
**Presentation:** 3
**Contribution:** 3
**Rating:** 8
**Confidence:** 4

**Summary:**

This paper introduces a novel reinforcement learning framework, Dynamic Reasoning Efficiency Reward (DRER), aimed at improving the quality of chain-of-thought reasoning in large language models. The key contributions are twofold. First, the DRER framework proposes a more nuanced reward signal that goes beyond simple answer correctness. It incorporates a "Reasoning Quality Reward" to credit reasoning steps that increase the likelihood of the correct answer, and a "Dynamic Length Advantage" to regulate the length of generated reasoning chains, which helps stabilize training. Second, the authors introduce LogicTree, a new benchmark dataset for deductive reasoning that is programmatically constructed. This dataset serves both as training data and as a tool for evaluating models in a controlled environment that focuses on logical structure over domain knowledge.

The core idea of rewarding reasoning chains that are demonstrably helpful is a significant step up from simply rewarding correct final answers. This is well-supported by strong experimental results showing that their method not only improves performance on their own benchmark but also generalizes to other established reasoning datasets. The ablation studies and analyses provide convincing evidence for the effectiveness of the different components of their proposed framework.

The LogicTree dataset is another major strength. By programmatically generating problems and controlling for logical depth, the authors provide a much-needed tool for disentangling pure reasoning ability from domain knowledge. While I have some questions about the generation process (see below), the dataset itself is a valuable resource for the research community.

Overall, the paper is well-written, the ideas are novel, and the empirical evaluation is thorough. The work opens up interesting avenues for future research on improving the reliability and transparency of reasoning in LLMs.

**Strengths:**

**Strong Points:**

*   **Novel Reward Framework:** The proposed DRER framework is a significant contribution. Moving beyond binary correctness to reward intermediate reasoning steps that are causally beneficial to the final answer is a promising direction for improving LLM reasoning.
*   **Practical Training Improvements:** The Dynamic Length Advantage is a practical and useful technique for controlling output length and stabilizing RL-based training of LLMs for reasoning tasks.
*   **Valuable New Dataset:** The LogicTree dataset is a strong contribution to the community. Its programmatic nature and focus on deductive reasoning provide a valuable tool for rigorously assessing and training the formal reasoning capabilities of models, separate from their factual knowledge.
*   **Strong Empirical Results:** The paper presents compelling experimental results. The proposed method shows significant improvements in accuracy and logical consistency over baselines. The demonstration of generalization to other logical and mathematical reasoning benchmarks is also a key strength.
*   **Insightful Analysis:** The analysis of how CoT reasoning impacts the model's predictions (e.g., when it is most effective at correcting a wrong answer) provides valuable insights into the behavior of these models.

**Weaknesses:**

**Weak Points:**

*   **Clarity on Dataset Generation:** The paper could benefit from a more detailed explanation of the lexicalization process used to create the LogicTree dataset. It is not entirely clear how the abstract logical rule trees are translated into natural language.
*   **Potential for Linguistic Ambiguity:** The process of converting logical forms to natural language is fraught with potential ambiguities (e.g., scalar implicatures, quantifier scope). The paper does not discuss whether these issues are addressed in the dataset generation process.
*   **Lack of Human Validation:** There is no mention of human validation for the LogicTree dataset. It would be valuable to know if human subjects consistently interpret the natural language problems in a way that aligns with the intended underlying logical structure.
*   **Minor Errors:** There are a few minor errors in the paper, such as a missing citation for a benchmark and some incorrect terminology for logical rules in a table.

**Questions:**

1.  Could you please provide more detail on the lexicalization process for the LogicTree dataset? Specifically, how are the abstract logical rules and entities mapped to natural language sentences?
2.  How do you ensure that the generated natural language statements are not subject to linguistic ambiguities, such as scalar implicatures (e.g., the interpretation of "or")? Have you considered or tested for such effects?
3.  Have you conducted any human evaluation studies on the LogicTree dataset? It would be very helpful to see data on how consistently human participants interpret the problems and whether their interpretation aligns with the intended logical form.

The following are some minor suggestions for improvement:

*   Please add the missing reference for the "Math500 benchmark" on page 4.
*   In Table 1, the names of the logical rules appear to be incorrect. For example, what is labeled "Conjunction introduction" seems to be the "Constructive Dilemma," and "conjunction elimination" appears to be "proof by cases." Please double-check this terminology.
*   In Figure 2, the "question" is technically a statement to be evaluated for its truth value. It might be clearer to label it as "Statement" or "Claim" to avoid confusion.

---

> ### Author Response · Authors · 2025-11-21
> **Responses for Reviewer Vyr9 on Weak1&Question1**
>
> Thank you for your professional and insightful comments. We sincerely appreciate your positive assessment of our work. Following your feedback, we have added detailed clarifications, supplemental validation experiments, and corresponding revisions throughout the paper. The code and dataset for this project will be fully released and continuously maintained. We hope the following responses adequately address your concerns, and we warmly welcome any further discussion.
>
> Below we provide point-by-point responses to your comments:
>
> **W1**: *Clarity on Dataset Generation: The paper could benefit from a more detailed explanation of the lexicalization process used to create the LogicTree dataset. It is not entirely clear how the abstract logical rule trees are translated into natural language.*
>
> **Q1**: *Could you please provide more detail on the lexicalization process for the LogicTree dataset? Specifically, how are the abstract logical rules and entities mapped to natural language sentences?*
>
> **A1**: Thank you very much for your recognition of the LogicTree dataset and for your interest in its construction. Due to space limitations, the main text indeed does not include a sufficiently detailed explanation of the dataset generation process. More details are provided in **Appendix A.2** and **Figure 2**.
>
> To clarify the lexicalization process, we would first restate the construction of the LogicTree itself. From an implementation perspective, the pipeline works as follows: all atomic propositions are sampled from a filtered subset of the *AscentKB* corpus. These atomic propositions are then combined using predefined propositional connectives (four types, aka sentential connectives) to form compound propositions that serve as premises or conclusions in the reasoning process.
>
> When generating a logical tree, we first sample a reasoning rule and use its conclusion as the root node. Then, in a bottom-up manner, at each iteration we:
>
> 1. sample a deductive rule,
> 2. treat the current node as the conclusion under that rule, and
> 3. generate its premise nodes according to the rule’s sentential logic specification.
>
> Next, we randomly select one (or several) of these premises as the new target conclusion for the next iteration, delete that node, and create two new nodes to expand the reasoning depth or breadth. Repeating this procedure produces LogicTrees of controllable depth and width.
>
> If we examine the seven deductive rules in terms of the propositional connectives involved — NOT, AND, OR, and IMPLICATION — only NOT, OR, and IMPLICATION require explicit lexicalization. AND does not require special handling, as two true propositions can simply be listed sequentially. The OR connective is mapped through the *Disjunction* templates defined in `sentential.json`, while IMPLICATION is mapped through the *Conditional* templates defined in the same file.
>
> Thus, the lexicalization process only involves:
>
> 1. sampling atomic propositions, and
> 2. translating three types of sentential connectives (acutally four in code, as our implementation distinguishes `Statement`—true propositions—from `NOT` negations).
>
> Each connective corresponds to multiple predefined templates stored in `sentential.json`, ensuring adequate linguistic variety.
>
> Sincerely hope my explanation is clear enough to answer your questions.

---

> ### Author Response · Authors · 2025-11-21
> **Responses for Reviewer Vyr9 on Weak2&Question2**
>
> **W2**: *Potential for Linguistic Ambiguity: The process of converting logical forms to natural language is fraught with potential ambiguities (e.g., scalar implicatures, quantifier scope). The paper does not discuss whether these issues are addressed in the dataset generation process.*
>
> **Q2**: *How do you ensure that the generated natural language statements are not subject to linguistic ambiguities, such as scalar implicatures (e.g., the interpretation of "or")? Have you considered or tested for such effects?*
>
> **A2**： Thank you for raising this important point. When converting formal sentential logic into natural language, it is indeed possible to introduce linguistic ambiguities, such as amphiboly, that may interfere with the clarity of the underlying logical structure. As mentioned in our previous response, we designed the data generation pipeline specifically to minimize such risks.
>
> First, we restrict the task to **Propositional Logic** rather than first-order logic, thereby avoiding quantifiers and the semantic ambiguity they may introduce. Our reasoning rules operate strictly over each propositions, and do not depend on semantic relationships between them.
>
> Second, to ensure the reliability of the lexicalization process, we performed rigorous cleaning of the AscentKB corpus. All atomic propositions are required to be declarative sentences without ambiguous references, without conjunctions, without subordinate clauses, and without imperative or inverted constructions. This guarantees that each atomic proposition can be mapped into logical form unambiguously.
>
> Third, we predefined the set of usable propositional connectives and their natural-language realizations in the `sentential.json` template file. This ensures that the natural-language rendering of compound propositions is both structurally correct and semantically precise.
>
> Extracting reliable logical structures from unstructured LLM-generated text is already difficult without leveraging structured resources such as knowledge graphs. To ensure that our analysis of model reasoning remains accurate and trustworthy, the current work therefore focuses on systematic evaluation within a primitive logical system. Extending this framework to support higher-order logical reasoning in LLMs is an important direction that we plan to explore in future work.

---

> ### Author Response · Authors · 2025-11-21
> **Responses for Reviewer Vyr9 on Weak3&Question3**
>
> **W3**: *Lack of Human Validation: There is no mention of human validation for the LogicTree dataset. It would be valuable to know if human subjects consistently interpret the natural language problems in a way that aligns with the intended underlying logical structure.*
>
> **Q3**: *Have you conducted any human evaluation studies on the LogicTree dataset? It would be very helpful to see data on how consistently human participants interpret the problems and whether their interpretation aligns with the intended logical form.*
>
> **A3**: This is an excellent and highly reasonable concern. If linguistic ambiguities were introduced during the natural-language generation of the problems, then—even if the underlying logical tree is perfectly valid—the resulting questions could become unsolvable or logically unclear. Incorporating human validation is indeed valuable for verifying dataset quality and increasing the credibility of the experimental results.
>
> As described in the previous answer, we carefully designed multiple safeguards in our data construction pipeline to ensure that the natural-language statements faithfully reflect the underlying logical structures. Earlier versions of the dataset were manually inspected in detail, and we have now conducted a more extensive human evaluation with additional volunteers. The updated evaluation results are summarized in the table below.
>
> We recruited 15 graduate students who had received systematic training in mathematical logic or introductory logic courses. For each difficulty level in the test set, we sampled 20 questions and asked the participants to answer them. We report the average accuracy across all participants as the reference score for each level.
>
> **Table. 19**: Comparison of LLM and Human accuracy on LogicTree across various logical depth
> | Model              | 1    | 2    | 3    | 4    | 5    | 6    | 7    | 8    | Avg. |
> |--------------------|------|------|------|------|------|------|------|------|------|
> | Qwen3-235B-A22B    | 0.96 | 0.83 | 0.66 | 0.71 | 0.46 | 0.32 | 0.25 | 0.07 | 0.53 |
> | Deepseek-R1        | 0.85 | 0.76 | 0.61 | 0.47 | 0.36 | 0.18 | 0.19 | 0.07 | 0.44 |
> | Claude-3.7-Sonnet  | 0.76 | 0.67 | 0.21 | 0.10 | 0.07 | 0.02 | 0.02 | 0.00 | 0.23 |
> | GPT-o4-mini        | 0.74 | 0.64 | 0.25 | 0.20 | 0.10 | 0.06 | 0.05 | 0.02 | 0.26 |
> | DAPO+DRER (Ours)   | 0.90 | 0.83 | 0.76 | 0.59 | 0.67 | 0.45 | 0.31 | 0.31 | 0.60 |
> | **Human**          | 1.00 | 1.00 | 0.98 | 0.93 | 0.85 | —    | —    | —    | —    |
>
> The results show that for problems of simple to moderate difficulty (reasoning depth 1–5), human participants consistently identified the implicit logical rules and produced correct answers. For deeper reasoning levels (6–8), although the problems remain theoretically solvable, the context length can exceed 1k tokens, making manual step-by-step deduction extremely tedious and error-prone. For this reason, depth-6–8 questions were excluded from human testing.
>
> Importantly, while modern LLMs can easily apply individual deductive rules in isolation, our experiments reveal that they struggle with **long-horizon reasoning consistency and reliability**. This is likely due to attention disruption or hallucination effects over long CoT sequences. In contrast, the proposed DRER module, when integrated into the RLVR framework for enhancing LLM reasoning, demonstrates a cost-balanced mechanism that improves both the effectiveness and reliability of chain-of-thought reasoning.

---

> ### Author Response · Authors · 2025-11-21
> **Responses for Reviewer Vyr9 on Weak4&Question4**
>
> **W4**: *Minor Errors: There are a few minor errors in the paper, such as a missing citation for a benchmark and some incorrect terminology for logical rules in a table.*
>
> **Q4**: *The following are some minor suggestions for improvement:
> Please add the missing reference for the "Math500 benchmark" on page 4.
> In Table 1, the names of the logical rules appear to be incorrect. For example, what is labeled "Conjunction introduction" seems to be the "Constructive Dilemma," and "conjunction elimination" appears to be "proof by cases." Please double-check this terminology.
> In Figure 2, the "question" is technically a statement to be evaluated for its truth value. It might be clearer to label it as "Statement" or "Claim" to avoid confusion.*
>
> **A4**: Thank you for pointing out these textual errors. We have added the missing citation for the Math500 dataset:
>
> > Dan Hendrycks, Collin Burns, Saurav Kadavath, Akul Arora, Steven Basart, Eric Tang, Dawn Song, and Jacob Steinhardt. Measuring mathematical problem solving with the MATH dataset. In Proceedings of the NeurIPS Track on Datasets and Benchmarks, 1, 2021.
>
> The logical rule names in **Table. 1** were indeed mislabeled. We have now corrected them according to the terminology used in **Figure. 2** and **Table. 9**.
>
> As you rightly pointed out, the term “question” in **Figure. 2** is technically a “statement whose truth value needs to be determined.” In our dataset, this field is internally named “question” for ease of use by the broader research community. After careful consideration, we have revised the figure and caption to use the more precise term “statement to be determined” to ensure clarity and rigor in the paper.
>
> All corresponding corrections have been updated in the PDF. Thank you again for your careful reading and valuable suggestions!

---

> ### Author Response · Authors · 2025-11-27
> **Response to Reviewer Vyr9**
>
> Dear Reviewer Vyr9,
>
> Thank you again for taking the time to review our submission. We sincerely appreciate your constructive comments and the effort you put into evaluating our work. We hope that our responses have addressed your concerns, and we would be glad to provide any further clarification if needed. **We look forward to your additional comments and further discussion.**
>
> Best regards,
> Authors

---

### Official Review · Reviewer_cjaS · 2025-10-31

**Soundness:** 2
**Presentation:** 2
**Contribution:** 2
**Rating:** 4
**Confidence:** 2

**Summary:**

The paper introduces DRER (Dynamic Reasoning Efficiency Reward), a plug-and-play reinforcement learning (RL) framework designed to improve the quality and efficiency of Chain-of-Thought (CoT) reasoning in large language models (LLMs). Unlike conventional RL approaches that reward only final answer correctness, DRER incorporates:
1.	Reasoning Quality Reward: Measures whether the CoT tokens increase the model’s confidence in the correct answer by comparing log-likelihoods with and without CoT.
2.	Dynamic Length Advantage: Penalizes responses that are significantly longer or shorter than a validation-derived length threshold, promoting concise and stable reasoning.
To support training and evaluation, the authors also release LogicTree, a synthetically generated dataset of nested deductive reasoning problems with controllable depth, logical diversity, and intermediate sub-questions.
Experiments on LogicTree and other benchmarks show that DRER improves accuracy, logical consistency, confidence, and token efficiency over baselines like GRPO and DAPO.

**Strengths:**

1.	The Reasoning Quality Reward directly ties intermediate reasoning steps to answer confidence. The Dynamic Length Advantage may stabilize training and reduce verbosity.
2.	Analyses on various aspects of models’ reasoning capabilities.

**Weaknesses:**

1.	LogicTree focuses exclusively on propositional deductive reasoning. What about inductive, abductive, analogical, or higher-order reasoning?
2.	The paper compares DRER-augmented training against baselines (GRPO/DAPO). Readers would be interested in component-wise ablation study. For example: does the length regularization alone improve accuracy, or is it only effective when combined with the quality reward?
3.	The paper could consider varying the degree of distributional shift (e.g., by perturbing LogicTree rules, mixing domains, or introducing noise). It’s unclear whether DRER’s gains are robust to OOD conditions, or if they merely reflect superficial transfer within a narrow reasoning regime.
4.	There are some typos in the article, such as citations in lines 44-58.

**Questions:**

1.	Can DRER be extended to reasoning tasks where ground-truth answers are less well-defined?
2.	Does encouraging higher confidence in answers via CoT increase the risk of overconfident hallucinations when the premises are insufficient?

---

> ### Author Response · Authors · 2025-11-21
> **Responses for Reviewer cjaS on Weak1**
>
> Thank you for your valuable suggestions. We are very glad that you have recognized the method we proposed. We also welcome you to further discuss with us and look forward to your reply. The following is our response to the specific questions you raised:
>
> **W1**: *LogicTree focuses exclusively on propositional deductive reasoning. What about inductive, abductive, analogical, or higher-order reasoning?*
>
> **A1**: Thank you for raising important questions and suggestions regarding the LogicTree dataset.
>
> In both formal logic and real-world reasoning tasks, **deductive reasoning remains the most classical and widely used form of logical inference**. Our dataset is built upon well-established logical systems—namely **propositional logic** and **first-order logic (FOL)**—whose principles also serve as the theoretical foundation of mathematical logic. As a result, LogicTree naturally supports the development of LLMs’ abilities in **systematic mathematical problem solving and mathematical proof generation**, making it highly relevant to practical applications.
>
> At the same time, deductive logic is a *fully developed and rigorously formalized* system that can be faithfully implemented through deterministic programmatic generation. This allows us to construct datasets with precise logical structure and to analyze models’ reasoning behavior in a fine-grained and reliable manner.
>
> By contrast, **inductive reasoning**, **abductive reasoning**, and **analogical reasoning** correspond to distinct logical paradigms that inherently involve probabilistic semantics, fuzzy semantics, or other forms of uncertainty. These forms of reasoning cannot be cleanly represented using deterministic, Boolean-based formal systems, and therefore cannot be exhaustively or reliably generated by code in the same way as deductive logic.
>
> Furthermore, deductive reasoning rules extend naturally to **higher-order logic (HOL)**. Although HOL expands the representational power of first-order logic—e.g., adding quantification over predicates and functions—it remains within the framework of classical logic, and does *not* require new fundamental inference rules. Consequently, a dataset based on deductive logic such as LogicTree is sufficient for **rigorous and trustworthy evaluation of RL-enhanced LLMs** on classical and mathematical logic tasks, while providing the potential for **generalization to mathematical, symbolic, and coding-reasoning settings**.
>
> In summary, LogicTree is intentionally grounded in deductive logic because it:
>
> - provides a fully formalizable and verifiable reasoning environment,
> - aligns with the foundations of mathematical and classical logic,
> - supports systematic analysis of LLM reasoning behaviors, and
> - offers strong generalization potential to real-world reasoning domains.

---

> > ### Author Response · Authors · 2025-11-21
> > **Responses for Reviewer cjaS on Weak2**
> >
> > **W2**: *The paper compares DRER-augmented training against baselines (GRPO/DAPO). Readers would be interested in component-wise ablation study. For example: does the length regularization alone improve accuracy, or is it only effective when combined with the quality reward?*
> >
> > **A2**: Thank you for the suggestion. We have incorporated additional ablation analyses into the revised manuscript, now included in Appendix F. These experiments further examine the individual contributions of **RQR** and **DLA**, and we also provide the average response length curves from the DLA ablation study.
> >
> > In our training setup, we observed that **DLA alone yields a very minor improvement in accuracy**, while its main benefit lies in stabilizing training by mitigating the impact of extreme-length trajectories. The detailed experimental results are presented in **Table 19 of Appendix F**.

---

> ### Author Response · Authors · 2025-11-21
> **Responses for Reviewer cjaS on Weak3&4**
>
> **W3**: *The paper could consider varying the degree of distributional shift (e.g., by perturbing LogicTree rules, mixing domains, or introducing noise). It’s unclear whether DRER’s gains are robust to OOD conditions, or if they merely reflect superficial transfer within a narrow reasoning regime.*
>
> **A3**:Thank you for the helpful suggestion. Building on the original experiments, we have added new training results using **open-source reasoning datasets**, including *math data from SkyWork OR1-RL-Data* and *general reasoning data from WebInstruct*. We also expanded our evaluation to a wider range of benchmarks. The comparison between baseline algorithms and our method after 400 training steps is shown in **Table 2**.
>
> The experimental results demonstrate that:
>
> 1. **The gains from the DRER framework are not tied to any specific training dataset**, and the method shows strong generalization across both mathematical tasks and diverse reasoning benchmarks.
>
> 2. **Models trained on LogicTree also exhibit improvements on general mathematical benchmarks**, often outperforming baseline methods trained under the same settings.
>
> In addition, we randomly sampled CoT outputs from *all evaluated benchmarks* before and after DRER training, and conducted **GPT-5.1–based scoring** to assess reasoning quality. Detailed procedures and results can be found in **Appendix E**.
>
> **W4**: *There are some typos in the article, such as citations in lines 44-58.*
>
> **A4**:Thank you for the suggestion. We will carefully review and correct all minor issues, including the citation errors in lines 44–58. These issues have been fixed in the revised version of the manuscript.

---

> > ### Author Response · Authors · 2025-11-21
> > **Responses for Reviewer cjaS on Question1**
> >
> > **Q1**: *Can DRER be extended to reasoning tasks where ground-truth answers are less well-defined?*
> >
> > **A5**:Thank you for your question.
> >
> > Yes. DRER can naturally extend to reasoning tasks in which the ground-truth answer is uncertain. The Reasoning Quality Reward (RQR) in DRER is essentially a measure of how much the CoT tokens contribute to the model’s ability to infer correct answers, and assigns additional reward to CoT trajectories that increase this internal likelihood.
> >
> > From an information-theoretic perspective, this corresponds to the mutual information between the generated CoT and the model’s prediction of the correct answer. In other words, DRER encourages CoT that increases the model’s confidence in predicting correct outcomes.
> >
> > Formally, this can be written as:
> >
> > $$
> > I(X;Y) = H(Y) - H(Y\mid X)
> > $$
> >
> > or, conditioning on the model input \(x\), CoT \(z\), and a reference answer distribution $p^*$:
> >
> > $$
> > I(z; y^* \mid x)
> > = \mathbb{E}_{z, y^* \mid x}
> > \big[\log p(y^* \mid x, z) - \log p(y^* \mid x)\big].
> > $$
> >
> > Thus, DRER only requires a *reference signal* that reflects answer correctness, enabling comparison between two reasoning paths without relying on a single deterministic ground-truth answer. This reference can be derived in multiple ways.
> >
> > For open-ended questions, the “answer” is typically a distribution $p^*$.
> > DRER naturally generalizes as:
> >
> > $$
> > \Delta = \mathbb{E}_{y \sim p^\*}[\log p(y \mid s\_{\mathrm{cot}}) - \log p(y \mid s\_{\mathrm{nocot}})]
> > $$
> >
> > In open-ended RL reasoning tasks, reward models are often used to score answers, or TTRL-style self-voting methods are used to select pseudo-ground-truth candidates. Self-rewarding methods similarly rely on the model’s internal ability to evaluate multiple candidate responses. DRER is fully compatible with such approaches: it can use any high-quality answer distribution as the reference $p^*$, and compute how much the CoT tokens improve the likelihood of better answers—regardless of whether the answer is deterministic.
> >
> > Therefore, DRER can indeed be extended to open-ended reasoning tasks.
> > In this work, however, we mainly focus on RLVR-style tasks, where DRER provides a fine-grained CoT-based reward to improve reasoning quality.
> >
> > Thank you again for your thoughtful question. We hope this explanation fully addresses your concerns, and we greatly appreciate the opportunity to discuss this further.

---

> ### Author Response · Authors · 2025-11-21
> **Responses for Reviewer cjaS on Question2**
>
> **Q2**: *Does encouraging higher confidence in answers via CoT increase the risk of overconfident hallucinations when the premises are insufficient?*
>
> **A6**: We appreciate the reviewer’s insightful question regarding whether encouraging higher confidence through CoT may inadvertently increase overconfident hallucinations—particularly when the premises are insufficient or ambiguous.
>
> We address this concern in two parts:
> **(1) why DRER’s Reasoning Quality Reward (RQR) inherently avoids amplifying overconfidence**
>
> Although RQR compares likelihoods between CoT and no-CoT modes, it is *not* a confidence-amplification reward. Instead, it is a **ground-truth–anchored differential signal** that evaluates whether reasoning *improves correctness*, not whether it increases verbal confidence.
>
> RQR is defined as:
>
>  $$\Delta=\log p_{\theta}(y^{\*}\mid x,z_{\cot}) - \log p_{\theta}(y^{\*}\mid x,z_{\mathrm{nocot}}).$$
>
>
> Critically, the reward is computed on the **ground-truth token** $y^{*}$, not the model’s self-generated answer. Therefore:
>
> -  If CoT **fails to support reasoning toward the correct answer** (i.e., it does not increase the likelihood of predicting the ground-truth token $y^{*}$, then  RQR becomes **non-positive**, and such CoT trajectories are **penalized**.
> - Only when CoT provides reasoning that **helps predict the correct answer** does Δ become positive.
>
> This mechanism prevents CoT from being rewarded merely for sounding confident.
>
> **(2) empirical evidence validating that DRER reduces, rather than increases, hallucination-like behaviors in underdetermined scenarios.**
>
> To further address the reviewer’s concern and verify that RQR aligns with genuine reasoning quality—not merely confidence—we included additional external validation experiments.
>
> **Experiment A: External Evaluation of CoT Quality Using GPT-5.1**
>
> We use GPT-5.1 to evaluate the model-generated CoT trajectories along correctness, logical coherence, and necessity.
> Based on GPT-5.1 CoT scores, we bucket all examples and compute mean RQR. As shown in **Figure 11** (details in Appendix E.1):
>
> - **High-quality CoTs yield higher RQR**
> - **Low-quality CoTs receive significantly lower RQR**
> - **After DRER training, CoT-quality distribution shifts upward**, and RQR improves across all buckets
>
> This shows that **RQR aligns closely with external assessments of reasoning quality**.
>
> **Experiment B: Sensitivity of RQR under CoT perturbation**
>
> Across 2,000 evaluation samples:
>
> - **Original CoT**: Avg Δ = 0.29 (positive 73%)
> - **Shuffled CoT**: Avg Δ = 0.08 (positive 41%)
> - **Cross-question CoT**: Avg Δ = –0.34 (positive 7%)
>
> The monotonic pattern **Original > Shuffled > Cross** indicates RQR is highly sensitive to semantic relevance and structural coherence.
>
> ### **Experiment C: WR/RW/WW/RR behavior**
>
> - **WR (wrong → right):** largest Δ
> - **RW (right → wrong):** negative Δ
> - **WW/RR:** stable and consistent with expectations
>
> This confirms that Δ captures whether reasoning helps or harms correctness—not confidence.

---

> ### Author Response · Authors · 2025-11-27
> **Response to Reviewer cjaS**
>
> Dear Reviewer cjaS,
>
> Thank you again for taking the time to review our submission. We sincerely appreciate your constructive comments and the effort you put into evaluating our work. We hope that our responses have addressed your concerns, and we would be glad to provide any further clarification if needed. **We look forward to your additional comments and further discussion.**
>
> Best regards,
> Authors

---

### Official Review · Reviewer_T1yU · 2025-11-01

**Soundness:** 3
**Presentation:** 3
**Contribution:** 2
**Rating:** 4
**Confidence:** 5

**Summary:**

This paper proposes a new reinforcement learning framework for reasoning enhancement in large language models (LLMs).
The method introduces a Reasoning-Quality Reward (RQR) that measures the average log-likelihood difference between the model’s correct answer under reasoning (CoT) and direct answering (No-CoT) modes.
A positive difference implies that reasoning increases the model’s confidence in the correct answer, which is rewarded via a tanh-based scaling.
Additionally, the paper adds a Dynamic Length Advantage (DLA) term that regularizes overly long or short reasoning chains based on length statistics collected from validation rounds.
Experiments on a self-constructed LogicTree dataset and several reasoning benchmarks (AIME24, MMLU-redux, ProntoQA) show improved reasoning efficiency and robustness compared with GRPO and DAPO baselines.

**Strengths:**

1. Proposes a clear and interpretable framework to assess whether reasoning helps model confidence.

2. Combines reasoning-quality reward with a dynamic length regularization to stabilize training.

3. Experiments are controlled, fair, and internally consistent.

4. Writing and visualization are of high quality.

5. Provides an interesting diagnostic tool for analyzing reasoning effectiveness in LLMs.

**Weaknesses:**

1. The log-likelihood difference is a self-referential reward; it optimizes internal belief alignment rather than genuine reasoning improvement.
This is conceptually close to confidence calibration and may lead to reward hacking or “self-confirmation loops.”

2. Dynamic Length Advantage is highly similar to prior adaptive length penalties. The paper does not provide an ablation proving the unique benefit of its specific formulation.

3. The approach requires ground-truth answers and double forward passes, limiting scalability and applicability to open-ended reasoning tasks.

4. Gains on real reasoning benchmarks are modest; LogicTree is synthetic and may not reflect real reasoning complexity.

5. Related Work misses critical discussion of existing “self-consistency” or “critique-guided” RL frameworks (e.g., CFT, CRL, Self-Refine).

**Questions:**

1. The “Reasoning Quality Reward” is computed from the log-likelihood gap between reasoning and direct-answer modes.
Can you show that this signal tracks true reasoning correctness rather than just higher output confidence?
For instance, do higher rewards actually correspond to more logically valid reasoning chains?


2. Since the reward depends on two separate generations per example, it may fluctuate.
How stable is this log-likelihood difference during training?
Did you observe large variance across steps or seeds that could destabilize learning?


3. Have you checked whether the model’s reasoning becomes objectively better when judged by an external verifier or a stronger model (e.g., GPT-4 or a logic checker)?
That would help confirm the reward improves reasoning, not just internal consistency.


4. The length adjustment idea resembles prior work like Adaptive Length Penalty, What is genuinely new here, and how much does your version contribute beyond those methods?


5. How could it generalize to open-ended questions or dialogue, where no ground truth exists?

---

> ### Author Response · Authors · 2025-11-21
> **Responses for Reviewer T1yU on Weak1&Question1 (part1)**
>
> Thank you for your detailed and constructive suggestions. We have supplemented a large number of experiments and revised the paper based on your feedback, hoping to comprehensively demonstrate the effectiveness of our method. We also welcome further discussions with you and look forward to your reply. Your recognition is a great encouragement to our work.
> The following is our response to the specific questions you raised:
>
> **W1**: *The log-likelihood difference is a self-referential reward ... is conceptually close to confidence calibration and may lead to reward hacking or “self-confirmation loops.”*
>
> **Q1**: *The “Reasoning Quality Reward” is computed from the log-likelihood gap ... Can you show that this signal tracks true reasoning correctness rather than just higher output confidence?*
>
> **A1**: W1 and Q1 raise closely related concerns, so we address them together here.
>
> 1. **Clarification on whether the log-likelihood–based reward introduces self-referential risk**
>
>     We sincerely appreciate the reviewer’s insightful question regarding the potential “self-referential” nature of log-likelihood–based rewards. We fully agree with recent critique-guided RL work (e.g., CRL, CFT) that emphasizes a key principle: **process supervision should be grounded in verifiable**, **externally anchored signals** to avoid self-confirmation loops or confidence-amplification biases.
>
>     However, we would like to clarify that **RQR is specifically designed to avoid such self-referential issues**, even though it uses model likelihoods. This is because the logic of RQR is **anchored entirely on the ground-truth answer**, not the model’s own outputs.
>
>     The reward does not encourage higher “absolute confidence,” but measures **whether the CoT genuinely increases the model’s support for the correct answer**.
>
>     - **(1) RQR fundamentally differs from traditional confidence-based rewards**: RQR is defined as the difference between log-likelihoods under CoT vs. No-CoT, but crucially it is anchored on *the external correct token*:
>
>         $$\Delta=\log p_{\theta}(y^{\*}\mid x,z_{\cot}) - \log p_{\theta}(y^{\*}\mid x,z_{\mathrm{nocot}}).$$
>
>         Here, $y^*$ is the ground-truth answer token, not the model’s preferred output.
>         Therefore-even if the CoT appears more confident-if it does not genuinely increase the probability of the correct token compared with direct answering, the reward does not increase.
>
>     - **(2) Information-theoretic interpretation**: RQR corresponds to the sample estimate of the mutual information between CoT and the correct answer:
>
>         $$I(z;y^\*|x)=\mathbb{E}_{z,y^\*|x}[\log p(y^\*|x,z)-\log p(y^\*|x)]$$
>
>
>         This quantity measures the **information gain** contributed by CoT tokens toward predicting the correct answer.
>
>         This is fundamentally different from confidence-boosting or calibration methods: RQR is positive **only when the CoT provides additional, genuine information that helps predict the correct token**.
>
>         Thus, RQR encourages informative reasoning, not stable wording or self-generated confidence.
>
>     - **(3) RQR is a differential structure.** Because RQR measures a difference, non-reasoning factors—such as prompt format, linguistic fluency, or dataset difficulty—are largely filtered out.
>         These properties make RQR significantly less vulnerable to reward hacking or self-confirmation loops than typical “confidence-boosting” reward formulations.
>     - **(4) Complementarity with critique-guided RL**: While critique-based methods (e.g., CFT, CRL) supervise the content of critiques or judgments, RQR supervises the effect of reasoning tokens on correctness via a counterfactual comparison.
>
>         The two families are conceptually aligned but mechanistically different: RQR offers a **numerical, verifier-free, and lightweight form of process supervision**, making it complementary to critique-based RL.
>
> 2. **Additional external validation experiments**
>
>     To further address the reviewer’s concern and verify that RQR aligns with genuine reasoning quality—not merely confidence—we included additional external validation experiments.
>
>     **Experiment A: External Evaluation of CoT Quality Using GPT-5.1**
>
>     We first use GPT-5.1 to evaluate the model-generated CoT trajectories along three complementary dimensions: **correctness**, **logical coherence**, and **necessity**. These dimensions directly reflect the structural quality of the reasoning chain rather than surface fluency, making them a reliable external assessment signal.
>
>     Based on the overall CoT Score assigned by GPT-5.1, we bucket all examples and compute the mean RQR for each bucket, as shown in **Figure 11** (full experimental details are provided in **Appendix E.1)**.

---

> ### Author Response · Authors · 2025-11-21
> **Responses for Reviewer T1yU on Weak1&Question1 (part2)**
>
> 2. **Additional external validation experiments**
> [continue]
>
>     Three key trends emerge in experiment A:
>     - **(1) Higher-quality CoTs yield higher RQR values (a clear monotonic trend)**: High-scoring reasoning chains not only appear more frequently (bars) but also obtain substantially higher mean RQR (lines).
>
>     - **(2) Low-quality CoTs receive significantly lower RQR**: When the reasoning chain contains logical errors or unnecessary steps, its RQR decreases accordingly.
>
>     - **(3) After DRER training, the CoT-quality distribution shifts toward higher-score buckets, and RQR improves across all buckets**: This indicates that DRER does not merely amplify confidence but genuinely strengthens the logical validity of the reasoning process.
>
>     Together, these findings show that **RQR aligns closely with GPT-5.1’s external assessment of reasoning quality—higher rewards correspond to more logically valid and effective reasoning chains rather than superficial confidence.**
>
>     **Experiment B: Sensitivity Evaluation of RQR with CoT perturbation**
>
>     To evaluate the sensitivity of RQR to cot quality, we performed a CoT perturbation experiment on 2,000 evaluation samples.
>
>     For each problem, we constructed three types of reasoning chains:
>     - **Original CoT** (model-generated, coherent, and semantically relevant)
>     - **Shuffled CoT** (same content but sentence order is randomly permuted, damaging logical structure)
>     - **Cross-question CoT** (taken from another problem and thus semantically irrelevant)
>
>     We computed RQR for each. Results:
>     - **Original CoT**: Avg RQR = 0.29, positive 73%
>     - **Shuffled CoT**: Avg RQR = 0.08, positive 41%
>     - **Cross-question CoT**: Avg RQR = –0.34, positive 7%
>
>     The monotonic pattern **Original > Shuffled > Cross** is clear. Original CoTs substantially increase RQR; shuffled ones decrease it; unrelated CoTs yield negative RQR.
>
>     This demonstrates that **RQR is highly sensitive to semantic relevance and structural coherence**, and does not provide erroneous gains for irrelevant or broken reasoning chains. Experiment B therefore provides independent evidence that RQR reflects the **effective contribution** of reasoning steps.
>
>     **Experiment C: WR/RW/WW/RR behavioral analysis**
>
>     Preliminary analysis shows:
>     - **WR (wrong → right via CoT)**: Δ is largest
>     - **RW (right → wrong due to CoT)**: Δ is negative
>     - **WW/RR**: Δ is near zero or consistent with expected trends
>
>     This pattern confirms that Δ does not measure “confidence boosting,” but whether reasoning helps or harms correctness.
>
>     Together, these experiments show that RQR reliably tracks reasoning quality.
>
> **Additional observation: need for DLA**
>
> We observed that, in the early stages of training, a small number of samples with extreme-length CoTs can push Δ close to positive values, thereby affecting training stability. To address this issue, we introduced **DLA**, which filters out such extreme-length cases. Additional details are provided in our responses to Weakness 2 and Question 4.
>
> Thank you for the reviewer’s valuable comments.
>
> DLA is a length-regularization component we propose to **shield the training process from the destabilizing influence of extreme-length trajectories**. As discussed in the paper, we observed that a very small number of extremely long or short samples appearing in early RL steps can substantially affect the model’s learned response length. DLA is therefore introduced as a functional module within DRER to normalize these outlier samples so that **they cannot dominate the learning dynamics**.
>
> To the best of our knowledge, mainstream adaptive length penalties primarily focus on **improving reasoning efficiency**, typically by incorporating response length, problem difficulty, or similar factors into the reward. However, these approaches **cannot prevent extreme-length samples from harming training stability**.
>
> In contrast, DLA derives empirical length thresholds from validation-time statistics, and applies a logarithmic decay **directly at the advantage level** for samples whose lengths fall far outside these thresholds. This mechanism effectively stabilizes training without globally penalizing all responses.
>
> Therefore, DLA differs from prior adaptive length penalties in its **motivation**, **definition**, and **practical effect**.
>
> To more clearly illustrate the role of each component in DRER, we provide additional ablation studies in **Appendix F**. The results show that:
>
> 1. **DLA alone improves post-training accuracy**, indicating that suppressing extreme-length trajectories benefits learning;
> 2. **DLA provides significantly better stability of response length** throughout training compared to existing length-penalty methods.

---

> > ### Author Response · Authors · 2025-11-21
> > **Responses for Reviewer T1yU on Weak2&Question4**
> >
> > **W2**: *Dynamic Length Advantage is highly similar to prior adaptive length penalties. The paper does not provide an ablation proving the unique benefit of its specific formulation.*
> >
> > **Q4**: *The length adjustment idea resembles prior work like Adaptive Length Penalty, What is genuinely new here, and how much does your version contribute beyond those methods?*
> >
> > **A2**: Thank you for the reviewer’s valuable comments.
> >
> > DLA is a length-regularization component we propose to **shield the training process from the destabilizing influence of extreme-length trajectories**. As discussed in the paper, we observed that a very small number of extremely long or short samples appearing in early RL steps can substantially affect the model’s learned response length. DLA is therefore introduced as a functional module within DRER to normalize these outlier samples so that **they cannot dominate the learning dynamics**.
> >
> > To the best of our knowledge, mainstream adaptive length penalties primarily focus on **improving reasoning efficiency**, typically by incorporating response length, problem difficulty, or similar factors into the reward. However, these approaches **cannot prevent extreme-length samples from harming training stability**.
> >
> > In contrast, DLA derives empirical length thresholds from validation-time statistics, and applies a logarithmic decay **directly at the advantage level** for samples whose lengths fall far outside these thresholds. This mechanism effectively stabilizes training without globally penalizing all responses.
> >
> > Therefore, DLA differs from prior adaptive length penalties in its **motivation**, **definition**, and **practical effect**.
> >
> > To more clearly illustrate the role of each component in DRER, we provide additional ablation studies in **Appendix F**. The results show that:
> >
> > 1. **DLA alone improves post-training accuracy**, indicating that suppressing extreme-length trajectories benefits learning;
> > 2. **DLA provides significantly better stability of response length** throughout training compared to existing length-penalty methods.

---

> > > ### Author Response · Authors · 2025-11-21
> > > **Responses for Reviewer T1yU on Weak3&Question5**
> > >
> > > **W3**: *The approach requires ground-truth answers and double forward passes, limiting scalability and applicability to open-ended reasoning tasks.*
> > >
> > > **Q5**: *How could it generalize to open-ended questions or dialogue, where no ground truth exists?*
> > >
> > > **A3**: W3 and Q5 raise closely related concerns, so we address them together here.
> > >
> > > 1. We appreciate the reviewer’s insightful comments. DRER indeed requires two forward passes.
> > >
> > >     For clarity, this design directly corresponds to our motivation. One of our core questions is whether the RL-induced CoT genuinely helps the model solve reasoning problems **from the perspective of its internal probability distribution**. To measure this effect, we compute the *delta log-probability* at the answer positions under two different prefixes — with and without CoT. This requires two forward passes but allows us to precisely quantify the contribution of the generated reasoning to the model’s confidence, which is essential for computing our Reasoning Quality Reward.
> > >
> > >     From an algorithmic perspective, the double forward computation can be further optimized. For example:
> > >
> > >     - reusing the KV-cache to avoid recomputing the shared prefix
> > >     - or directly gathering the token probabilities at answer positions across different prefixes
> > >
> > >     both significantly reduce the overhead of the second forward pass.
> > >
> > >     In practice, however, the additional cost introduced by the double forward process is entirely manageable. Empirically, DRER increases the average per-step training time by only **16.4 seconds**, approximately **17.3%** compared to baseline algorithms. With the above optimizations enabled, the added latency can be largely reduced, which has negligible impact on training speed.
> > >
> > >     We will provide the optimized implementation in our public codebase in a future update.
> > >
> > > 2. DRER can naturally extend to reasoning tasks where the ground-truth answer is uncertain or not explicitly provided.
> > >
> > >     The RQR component in DRER — the Reasoning Quality Reward — essentially measures the contribution of CoT tokens to the model’s ability to infer correct answers, and assigns additional reward to CoT trajectories that lead to higher likelihood of correct inference.
> > >
> > >     As discussed earlier, from an information-theoretic perspective, this corresponds to a form of mutual information between the generated CoT and the model’s prediction of the correct answer. Conceptually, DRER requires a **reference signal** that reflects correctness, enabling comparison between two reasoning paths (with or without CoT), without relying on a single deterministic ground-truth answer. This signal can be derived in multiple ways.
> > >
> > >     For open-ended questions, the “answer” is typically a distribution $p^*$.
> > >     DRER can be naturally generalized as follows:
> > >
> > >     $$
> > >     \Delta = \mathbb{E}_{y \sim p^\*} \left[ \log p(y \mid s\_{\text{cot}}) - \log p(y \mid s\_{\text{nocot}}) \right].
> > >     $$
> > >
> > >     In open-ended RL reasoning tasks, reward models commonly score answers, or use mechanisms such as TTRL-style self-voting to select a pseudo ground-truth answer set. Self-reward approaches also rely on the model’s own ability to rank or evaluate answers. DRER naturally aligns with these paradigms, since it can use any high-quality answer distribution as a reference, computing how much the CoT tokens contribute to improving the likelihood of better answers — regardless of whether these tokens point to a single “correct” answer.
> > >
> > >     Therefore, DRER can be extended to open-ended reasoning tasks. However, the present work primarily focuses on RLVR-style problems, where DRER leverages CoT-conditioned likelihood comparisons to provide more fine-grained reasoning rewards.

---

> > > > ### Author Response · Authors · 2025-11-21
> > > > **Responses for Reviewer T1yU on Weak4**
> > > >
> > > > **W4**: *Gains on real reasoning benchmarks are modest; LogicTree is synthetic and may not reflect real reasoning complexity.*
> > > >
> > > > **A4**: Thank you for the reviewer’s valuable comments.
> > > >
> > > > LogicTree is a scalable synthetic deductive reasoning dataset. Because its reasoning depth is fully controllable, it provides a suitable environment for analyzing model reasoning behaviors. In our additional experiments, we included a broader set of evaluation benchmarks, and the results show that:
> > > >
> > > > 1. Models trained on LogicTree generally achieve larger improvements on logic-focused evaluation benchmarks that require deductive reasoning.
> > > > 2. They also exhibit moderate gains on mathematical generalization benchmarks.
> > > >
> > > > Furthermore, to demonstrate that the plug-and-play nature of DRER does **not** rely on any specific property of LogicTree, we added experiments where DRER is trained on open-source reasoning datasets. The results indicate that DRER continues to yield significant improvements over baseline methods, confirming that its effectiveness is not dependent on LogicTree.

---

> > > > > ### Author Response · Authors · 2025-11-21
> > > > > **Responses for Reviewer T1yU on Question3**
> > > > >
> > > > > **Q3**: *Have you checked whether the model’s reasoning becomes objectively better when judged by an external verifier or a stronger model (e.g., GPT-4 or a logic checker)? That would help confirm the reward improves reasoning, not just internal consistency.*
> > > > >
> > > > > **A5**: Thank you for the valuable comments. As shown in Experiment A in our response to Q1, we used GPT-5.1 to evaluate the model’s CoT responses along the dimensions of correctness, logical consistency, and necessity. We find that CoTs generated after DRER training receive higher scores than those from baseline models, and their corresponding RQR values also increase significantly. In addition, in our supplementary experiments we evaluated DRER on several widely used benchmarks. DRER exhibits generally consistent improvements over the baselines across these datasets, with some benchmarks showing particularly pronounced gains.

---

> > > > > > ### Author Response · Authors · 2025-11-21
> > > > > > **Responses for Reviewer T1yU on Question2**
> > > > > >
> > > > > > **Q2**: *Since the reward depends on two separate generations per example, it may fluctuate. How stable is this log-likelihood difference during training? Did you observe large variance across steps or seeds that could destabilize learning?*
> > > > > >
> > > > > > **A6**: Thank you for the valuable question. We agree that introducing a “CoT vs. No-CoT log-likelihood difference” as a reward does numerically increase the variance of the reward signal. To assess its effect, we monitored reward variance throughout training in a systematic manner.
> > > > > >
> > > > > > Across two training datasets, we observed that the mean reward standard deviation of DRER is indeed higher—by approximately 57%—compared with baselines such as GRPO and DAPO. However, this increase did not cause instability during training, for two key reasons:
> > > > > >
> > > > > > 1. As shown in **Figure. 4**, RQR exhibits a stable upward trend and clear convergence throughout training, without the oscillation, gradient explosion, or policy collapse commonly seen in RL instability cases.
> > > > > > 2. Results across multiple random seeds are highly consistent, with overlapping learning curves, indicating that the increased reward variance does not translate into optimization instability.
> > > > > >
> > > > > > In addition, we applied standard stabilization techniques widely used in RLHF/GRPO frameworks—such as reward normalization, value baselines, and KL regularization—which further ensured the robustness of the training process.
> > > > > >
> > > > > > Therefore, although RQR introduces a larger numerical reward variance, its counterfactual difference structure and standard stabilization techniques keep the overall training stable. We did not observe any performance degradation attributable to the increased variance.

---

> ### Author Response · Authors · 2025-11-25
> **Responses for Reviewer T1yU on Weak5**
>
> **W5**: *Related Work misses critical discussion of existing “self-consistency” or “critique-guided” RL frameworks (e.g., CFT, CRL, Self-Refine).*
>
> **A7**: Thank you very much for pointing out this important omission. We fully agree that self-consistency and critique-guided reasoning paradigms—such as Self-Refine, Reflexion, CRITIC, CFT, and CRL—constitute an essential line of research for improving LLM reasoning. These approaches are conceptually related to our work and help contextualize the broader landscape of process-level reasoning supervision.
>
> In the revised manuscript, we have added a dedicated subsection (“Related Reasoning Methods” in the appendix) that systematically supplements and discusses these methods. Specifically, we:
>
> - Summarize self-improvement methods (Self-Refine, Reflexion, CRITIC) and cite recent analyses showing that their iterative critique loops may be unstable or overly sensitive to surface-level linguistic features.
> - Introduce supervised critique-generation methods such as CFT and Critique-Guided Distillation, clarifying that these approaches supervise the *content* of critiques rather than evaluating the causal contribution of reasoning steps to the correctness of the final answer.
> - Contrast CRL with DRER: CRL optimizes the correctness of True/False judgments over candidate solutions, whereas DRER directly measures whether intermediate reasoning steps increase support for the correct answer through a CoT–NoCoT log-likelihood difference.
> - Systematically distinguish DRER from process-level RL methods (e.g., GRPO, DAPO, RLVR) and reward-model-based reasoning methods.
>
> Through these additions, we clarify that DRER occupies a conceptually distinct position. It does not rely on critique generation, critique supervision, or external judgment signals; instead, it evaluates the effectiveness of intermediate reasoning through a counterfactual, ground-truth–anchored likelihood-difference reward. We believe this expanded discussion more accurately situates DRER within the broader landscape of reasoning-enhancement methods and fully addresses the reviewer’s concerns.
>
> Once again, we sincerely appreciate the reviewer’s insightful suggestion. We have incorporated the relevant references and comparisons into the revised manuscript.

---

> ### Author Response · Authors · 2025-11-27
> **Response to Reviewer T1yU**
>
> Dear Reviewer T1yU,
>
> Thank you again for taking the time to review our submission. We sincerely appreciate your constructive comments and the effort you put into evaluating our work. We hope that our responses have addressed your concerns, and we would be glad to provide any further clarification if needed. **We look forward to your additional comments and further discussion.**
>
> Best regards,
> Authors

---

### Official Review · Reviewer_qz22 · 2025-11-01

**Soundness:** 3
**Presentation:** 3
**Contribution:** 2
**Rating:** 4
**Confidence:** 4

**Summary:**

The core contribution of the paper is the proposal of a process-based reward mechanism, which promotes better reasoning by giving more reward to effective intermediate reasoning steps (CoTs), rather than relying solely on rule-based evaluation of the final answers.

**Strengths:**

1. Good Research Question: Only RLVR may not be enough to supervise model for reasoning, so more reward on CoT has the potential to be better.

**Weaknesses:**

1. The main results of the paper **rely on the LogicTree training set constructed by the authors and are evaluated using its corresponding test set**, which imposes significant limitations.

It is unclear **why existing open-source reasoning datasets could not be adapted instead**？

Relying solely on LogicTree as the training set **does not sufficiently support the claimed “plug-and-play” capability**.

2. Results on Table 4 should be in the place of Table 1 because results from these benchmarks are more reliable on reflecting reasoning abilities and cared about by the community.

3. Typo: Line 190, missing citation

**Questions:**

1. What is the relationship between this method and your LogicTree dataset? Could this method be applied while leveraging existing open-source reasoning datasets, rather than being restricted to constructing LogicTree?

2. the main experimental results should be evaluated across a variety of commonly used logic/reasoning datasets and compared with more methods.

---

> ### Author Response · Authors · 2025-11-21
> **Responses for Reviewer qz22**
>
> Thank you for your detailed and constructive suggestions. We are very glad that you have recognized our research issues. We have supplemented a large number of experiments and revised the paper based on your feedback. We also welcome you to further discuss with us and look forward to your reply.
> The following is our response to the specific questions you raised:
>
> **W1**: *The main results of the paper rely on the LogicTree training set constructed by the authors and are evaluated using its corresponding test set, which imposes significant limitations.*
> *It is unclear why existing open-source reasoning datasets could not be adapted instead？
> Relying solely on LogicTree as the training set does not sufficiently support the claimed “plug-and-play” capability.*
> **Q1**: *What is the relationship between this method and your LogicTree dataset? Could this method be applied while leveraging existing open-source reasoning datasets, rather than being restricted to constructing LogicTree?*
>
> **A1**: We fully agree with the reviewer that demonstrating the effectiveness of a plug-and-play method solely on LogicTree is insufficient. To address this concern, we have added additional experiments where DRER is trained on two publicly available reasoning datasets *Skywork-OR1-RL-Data*[1] and *General-Reasoner*[2]. We also re-evaluated all models with DRER and baselines on a broader set of public benchmarks, including AIME24, MATH500, MMLU-Pro, and LogiQA2.0, and have updated the results in **Table 2**.
>
> The results show that:
>
> 1. DRER consistently outperforms baseline methods when trained on other open-source reasoning datasets, demonstrating that its effectiveness does not rely on LogicTree.
> 2. Models trained on open-source mathematical reasoning datasets tend to perform better on math-focused benchmarks, whereas models trained on LogicTree show larger gains on logic-focused benchmarks.
>
> To address the limitation that outcome-only reward signals cannot provide fine-grained supervision for the generated CoT, we introduce DRER, which evaluates CoT quality based on the model’s change in confidence on the **ground-truth answer token**. This allows DRER to assign higher rewards to CoT trajectories that genuinely help the model reach the correct answer.
>
> In contrast, many contemporary approaches that aim to improve logical reasoning typically rely on increasing the mathematical difficulty or coding depth of training data, which makes it difficult to control logical depth and inference difficulty. To more precisely evaluate logical reasoning ability, we introduce LogicTree dataset, which enables controlled analysis on pure deductive reasoning. Although LogicTree is also used as part of our training data, DRER itself is not dependent on this dataset.
>
> **W2**: *Results on Table 4 should be in the place of Table 1 because results from these benchmarks are more reliable on reflecting reasoning abilities and cared about by the community.*
>
> **Q2**: *the main experimental results should be evaluated across a variety of commonly used logic/reasoning datasets and compared with more methods.*
>
> **A2**: Thank you for pointing out the limitations of our original experiments. We have now incorporated the additional experimental results, including training on open-source reasoning datasets, evaluations on more mainstream benchmarks, and comparisons with additional baselines (see our response to Q1) into **Table 2**.
>
> **W3**: *Typo: Line 190, missing citation.*
>
> **A3**: Thank you for pointing out this typo. We have carefully reviewed and corrected all minor issues, including the citation error at line 190. The revised version has already fixed this problem.
>
> [1] Skywork Open Reasoner 1 Technical Report, He, Jujie, et al., 2025.
>
> [2] General-reasoner: Advancing llm reasoning across all domains, Ma, Xueguang, et al., 2025

---

> ### Author Response · Authors · 2025-11-27
> **Response to Reviewer qz22**
>
> Dear Reviewer qz22,
>
> Thank you again for taking the time to review our submission. We sincerely appreciate your constructive comments and the effort you put into evaluating our work. We hope that our responses have addressed your concerns, and we would be glad to provide any further clarification if needed. **We look forward to your additional comments and further discussion.**
>
> Best regards,
> Authors

---

### Author Response · Authors · 2025-12-03
**Summary of Rebuttal**

We sincerely thank all reviewers for their constructive comments and for recognizing the contributions of our work. Below we summarize the acknowledged strengths, the major concerns, and how we have fully addressed them with substantial new experiments and clarifications.

---

###  **Contributions**
- We propose **DRER**, a **plug-and-play RL framework** that provides process-level supervision for CoT reasoning by rewarding truly helpful reasoning steps through a ground-truth–anchored likelihood-difference signal, along with Dynamic Length Advantage to stabilize reasoning efficiency.
- We introduce **LogicTree**, a programmatically generated dataset enabling fine-grained deductive reasoning evaluation with controllable complexity.

---

### **Reviewer-Recognized Strengths**

- **Clear and well-motivated formulation**: Reviewers agree that rewarding intermediate reasoning steps provides a sound and meaningful complement to RL-from-outcome training, addressing the limitations of purely final-answer supervision.
- **Strong and insightful empirical support**: The method consistently improves both accuracy and reasoning behavior, while analyses offer interpretable insight into how DRER helps correct model's CoT behaviour.
- **Valuable dataset contribution**: LogicTree is recognized as a useful resource that isolates deductive reasoning ability and supports systematic evaluation beyond domain knowledge.

---

### **Reviewer Major Concerns**

- **Whether DRER relies on LogicTree and lacks evidence of plug-and-play generalization** (Reviewer qz22):

  We trained DRER additionally on mixed general reasoning data (Skywork-OR1-RL-Data & General-Reasoner) and evaluated on AIME24, Math500, MMLU-Pro, LogiQA2.0, GPQA,  TheoremQA, LogiQA2.0 and ZebraLogic— demonstrating clear plug-and-play generalization across domains.

- **Whether RQR is self-referential and may optimize confidence rather than true reasoning** (Reviewer T1yU, cjaS)：

  We performed three external validations (GPT-5.1 scoring, CoT perturbation, WR/RW/RR/WW analysis), proving that RQR rewards logically beneficial reasoning rather than confidence amplification, and **DRER significantly improves the reasoning quality after training**.

- **Lack of independent ablation analysis verifying each component's contribution within DRER** (Reviewer cjaS):

  We added component-wise ablations showing **both** RQR and DLA provide distinct improvements in accuracy and training stability.

- **Dataset clarity, lexicalization, linguistic ambiguity, and lack of human validation** (Reviewer Vyr9):

  We added full lexicalization specification, semantic filtering process, and human evaluation confirming natural-language interpretations align with intended logical structures.

---

### **Final Remarks**
All concerns have been **fully addressed** with new experiments and clearer explanations.

We sincerely appreciate the reviewers’ insightful feedback and the AC’s efforts in evaluating this strengthened submission.

Best regards,

Authors

---

### Author Response · Authors · 2025-12-03
**Clarification Note for Summary**

Dear Area Chair,

Due to the early termination of the discussion phase, some of the key clarifications and additional experiments we proposed did not receive reviewer responses.  **To assist evaluation, we summarise the essential points of rebuttal, showing that all major concerns have been fully addressed.**

All improvements and new experimental results have been fully updated in the revised PDF.

**We sincerely appreciate your time and consideration.**

Best regards,

Authors

---

### Meta-Review · Area_Chair_oeVM · 2026-01-10

**Summary:**

The paper proposes DRER, a reinforcement-learning framework that rewards chain-of-thought steps if they increase the model’s likelihood on the correct answer (Reasoning Quality Reward) and regularizes reasoning length via Dynamic Length Advantage. Experiments use the authors’ LogicTree dataset and several public benchmarks. The rebuttal adds results trained on other datasets, more baselines, ablations, small human/LLM judge checks, and clarifies some presentation issues.

Pros
1. Clear motivation for process-level rewards beyond final-answer supervision.
2. Interpretable formulation (Δ log-likelihood with/without CoT) and a practical length stabilizer.
3. Rebuttal broadens experiments and fixes some clarity/typo issues.
4. Analyses of CoT behavior and some external judging provide useful diagnostics.

Cons
1. Limited novelty: the reward is closely related to confidence calibration/self-consistency signals; DLA resembles prior adaptive length penalties. Formal distinctions and theory are thin.
2. Evidence still hinges on a synthetic, author-curated dataset; the “plug-and-play” claim remains under-substantiated despite added runs. Gains on widely used external benchmarks are modest and not consistently superior across stronger alternatives.
3. Self-referential risk and reward hacking concerns are not fully resolved; external verification relies on small-scale LLM/human checks.
4. Practicality constraints: requires ground-truth answers and double forwards; scalability to open-ended settings is mostly speculative.

The idea is promising and the engineering is careful, but the incremental nature of the contributions, dependence on synthetic data, modest external gains, and unresolved concerns about generality and verification make the case below the bar this round. A stronger revision should (i) deliver standardized, budget-controlled comparisons on established reasoning suites with clearer wins; (ii) provide larger-scale human validation and clearer linguistic controls for LogicTree; (iii) add theory or stronger empirical evidence separating DRER from confidence-based/self-consistency baselines; and (iv) demonstrate applicability without ground-truth answers or heavy double-pass overhead.

**Reviewer Concerns:**

1. Added training on public datasets (Skywork-OR1-RL-Data, General-Reasoner) and broader evals (AIME24, MATH500, MMLU-Pro, LogiQA2.0); updated baselines.
2. Clarified DRER is not tied to LogicTree; fixed missing citation.
4. Self-referential risk: grounded on ground-truth token; provided info-theoretic view.
5. External validation: GPT-5.1 judging, CoT-perturbation sensitivity, WR/RW diagnostics.
6. DLA: ablations added; rationale and behavior curves provided.
7. Stability/variance: tracked variance, seeds, and convergence; reported ~17% step-time overhead with optimization notes.
8. Related work expanded (self-consistency/critique-guided RL, RM-based methods).

**Reviewer Scores:**

N/A

---

### Decision · Program_Chairs · 2026-01-26

Reject